# MLUCM BEP+BEM: An offline one-dimensional Multi-Layer Urban Canopy Model based on the BEP+BEM scheme

Gianluca PAPPACCOGLI[(1)], Andrea Zonato[(2)], Alberto Martilli[(3)], Riccardo Buccolieri[(1)], Piero Lionello[(1)]

[1]Dipartimento di Scienze e Tecnologie Biologiche ed Ambientali, University of Salento, Lecce, 73100, Italy
[2]Royal Netherlands Meteorological Institute (KNMI), De Bilt, The Netherlands
[3]Atmospheric Modelling Unit, Environmental Department, CIEMAT, 28040 Madrid, Spain

*Correspondence to*: Gianluca Pappaccogli (gianluca.pappaccogli@unisalento.it)

**Abstract.** The MLUCM BEP+BEM model advances urban microclimate modelling by combining a multi-layer canopy approach with Building Effect Parameterization (BEP) and a Building Energy Model (BEM). It includes updated turbulent length scales and eddy diffusivity coefficients that account for atmospheric stability, along with explicit representation of urban vegetation, such as street trees and green spaces. The model runs offline with low computational demands, making it suitable for standalone use, integration with climate projections, and long-term simulations to evaluate emission scenarios and adaptation strategies. Validation against data from the Urban-PLUMBER project at a suburban site in Preston (Melbourne, Australia) demonstrates that MLUCM BEP+BEM reliably reproduces shortwave ($SW_{up}$) and longwave ($LW_{up}$) radiation, as well as latent ($Q_{le}$), sensible ($Q_h$), and momentum ($Q_{tau}$) fluxes. Its overall performance is on par with, and in several cases surpasses, that of other established urban models with particularly notable improvements in the simulation of momentum flux ($Q_{tau}$). Some refinements are still needed, particularly in modelling tree-soil moisture dynamics to reduce surface energy budget imbalances. Thanks to its flexibility and efficiency, MLUCM BEP+BEM is well-suited for assessing urban overheating, building energy demand, and the effectiveness of mitigation strategies such as green roofs, cool materials, and photovoltaic systems under various future climate scenarios.

## 1 Introduction

Urban environments are inherently complex, shaped by physical and anthropogenic factors (Grimmond et al., 2010), and significantly influence local climate and meteorological conditions (Britter and Hanna, 2003; Oke et al., 2017). Accurately modelling these interactions is essential to address urban challenges such as overheating, air pollution, and energy consumption (Mills, 2007). The urban canopy layer's atmospheric processes are particularly intricate due to the morphology of buildings and vegetation, which interact with mesoscale processes (Santiago and Martilli, 2010; Krayenhoff et al., 2020). Mesoscale models must capture entire cities and their surrounding areas, accounting for urban-induced effects. However, computational

costs often limit model resolution, necessitating urban canopy parameterizations (UCPs) to accurately represent urban effects and their feedback on regional climates (Best, 2005; Martilli, 2007). UCPs serve to balance detailed urban effects representation with computational efficiency (Santiago and Martilli, 2010), addressing both dynamic effects and heat exchanges between surfaces and the atmosphere through various parameterization schemes. Significant advancements in

process-based models over the past two decades have enabled improved prediction of time-averaged micrometeorological effects within urban canopies (e.g., Masson, 2000; Kusaka et al., 2001; Martilli et al., 2002).

Urban canopy models (UCMs) offer distinct advantages, particularly in explicitly representing building geometry, radiative interactions, and surface-specific energy exchanges (Masson, 2006). These models account for heat transfer through conduction, convection, and radiation, as well as the drag induced by urban surfaces. UCMs can be categorized into single-

layer and multi-layer models. Single-layer models (e.g. Masson, 2000; Kusaka et al., 2001) represent the urban canopy as a single atmospheric layer, providing averaged estimates for temperature, wind speed, and humidity across the urban volume. In contrast, multi-layer models (e.g., Martilli et al., 2002) divide the canopy into vertical layers, offering a more detailed vertical representation of urban physics, including the variation in building and vegetation heights and improved predictions of street-level climate and pollutant dispersion. Some UCMs incorporate building energy models (Salamanca et al., 2010),

while others include urban vegetation (Dupont et al., 2004; Krayenhoff et al., 2020). The Building Effect Parameterization (BEP) scheme, coupled with the Building Energy Model (BEM), was integrated into the Weather Research and Forecasting (WRF) model starting from version 3.2 (Martilli et al., 2002; Salamanca et al., 2010). BEM simulates internal building thermal dynamics, including heat transfer through walls and windows and HVAC (Heating, Ventilation, and Air Conditioning) operations. This multi-layer scheme improves the accuracy of urban energy flux simulations by integrating building-specific

characteristics, such as insulation and heating/cooling systems, with atmospheric models (Pappaccogli et al., 2020, 2021). Since 2021, Zonato et al. have introduced additional parameterizations for green roofs, photovoltaic panels, and urban material permeability, along with the ability to estimate thermal comfort using the Universal Thermal Climate Index (UTCI) (Martilli et al., 2024). These developments focus on assessing the urban overheating effect and evaluating mitigation strategies for reducing urban temperatures and building energy consumption. The BEP+BEM model has demonstrated its effectiveness in

studying meteorological impacts on building energy use and developing adaptation strategies for optimizing energy consumption, particularly in the context of climate change and extreme weather events. Recent studies have also highlighted its application in simulating pollutant dispersion at the urban scale (Martilli et al., 2021; Martilli et al., 2022). However, the complexity of these schemes, coupled with mesoscale models, necessitates significant computational resources, limiting their use in long-term climate simulations and real-time forecasting.

This work focuses on developing MLUCM (Multi Level Urban Canopy Model), which is an offline 1-Dimensional version of BEP+BEM to reduce computational demands, making it suitable for extended simulations and novel applications. The key function of this model is to bridge the mesoscale and microscale phenomena occurring in the planetary boundary layer and in the canopy of urbanized environments with limited vegetation cover, accounting for exchanges and feedback between different scales and processes. Moreover, it makes the code development more straightforward and faster for urban climate modelers.

MLUCM BEP+BEM, incorporates the vertical turbulent diffusion scheme of Santiago and Martilli (2010) with the Building Effect Parameterization (BEP, Martilli et al., 2002) and the Building Energy Model (BEM, Salamanca et al., 2009). In this study we describe the MLUCM BEP+BEM and its validation using data from the Urban-PLUMBER project, which assessed surface-atmosphere fluxes at a suburban site in Preston, Melbourne, over a 16-month period (Lipson et al., 2024). Participation in intercomparison projects and the use of web-based platforms, such as modelevaluation.org, have been crucial in enhancing

model performance, establishing benchmarks, and offering training opportunities for new users. Further details on the proposed model and validation are given in the following sections.

The structure of the paper is as follows: Section 2 outlines the theoretical framework of the MLUCM BEP+BEM model and its recent advancements. Section 3 details the real-data case study utilized for model validation and explains the simulation setup employed in this research. Section 4 presents the results, emphasizing sensitivity analyses and comparative evaluations.

Finally, Section 5 concludes the study and discusses its implications for future research.

## 2 Section

The proposed BEP-BEM model is based on the Building Effect Parameterization (BEP) scheme by Martilli et al. (2002) and the Building Energy Model (BEM) by Salamanca et al. (2010). This model calculates source and sink terms induced by ground surfaces, building roofs, and walls, which are then used in vertical diffusion equations for horizontal wind, turbulent kinetic

energy, potential temperature, and specific humidity. Coupled to BEP-BEM is the one-dimensional (1-D) vertical turbulent diffusion model based on Santiago and Martilli (2010), with updates to turbulent length scales for dissipation and eddy coefficients, considering atmospheric stability, inspired to the Bougeault and Lacarrere (1989) turbulence scheme. This inclusion makes MLUCM BEP-BEM a comprehensive offline model that requires only atmospheric forcing at its top boundary (Figure 1), which can be provided from global or mesoscale models at a level above the roughness sublayer. This design allows

a one-way coupling of MLUCM BEP-BEM with an atmospheric model that allows MLUCM BEP-BEM to be run offline, without a significant penalty in computational time. Additionally, green areas and street trees are included based on Zonato et al. (2021) and Stone et al. (2021), respectively. The individual components of BEP-BEM and their respective updates are detailed below.

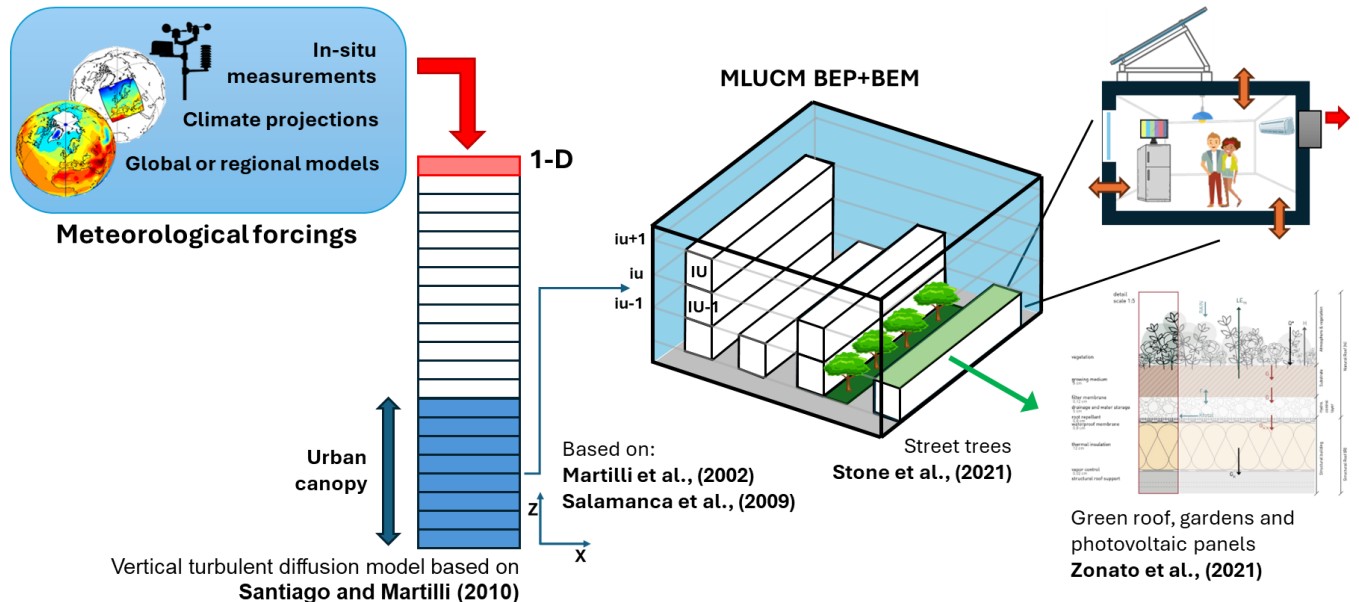


Figure 1: Flowchart of the MLUCM BEP+BEM model setup, illustrating the meteorological forcings applied at the top boundary, the integration of a one-dimensional turbulent diffusion scheme, and key model components, including the schematic representation of BEM processes and green roof elements, modified from Zonato et al., (2021).

## 2.1 Model description


The proposed MLUCM BEP+BEM is based on a 1-D column model with $k - l$ turbulence closure, as proposed by Santiago and Martilli (2010). The conservation equation for the horizontal Reynolds-averaged velocity components $u, v$, assuming incompressibility and horizontal homogeneity (and thus mean vertical velocity $w$ equal to zero), is:

$$\frac{\partial \rho \langle \overline{u} \rangle}{\partial t} = -\frac{\partial \rho \langle \overline{u' w'} \rangle}{\partial z} + \rho D_u \quad ; \quad \frac{\partial \rho \langle \overline{v} \rangle}{\partial t} = -\frac{\partial \rho \langle \overline{v' w'} \rangle}{\partial z} + \rho D_v \quad (1)$$


where $u$ is the horizontal and $w$ the vertical component of velocity, $\rho$ is air density, and $\langle \rangle$ represents horizontal spatial average, overbar time or ensemble average, and ′ departure of instantaneous values from the time average. Accordingly, the first term on the right-hand side of Equation 1 represents the vertical gradient of the time-averaged turbulent momentum flux. The second term represents the drag induced by the buildings, parameterized as $D_u = -S(z)C_D|<\overline{u}>|<\overline{u}>, D_v = -S(z)C_D|<\overline{v}>|<\overline{v}>$ where $S(z)$ is the density of vertical surfaces. To estimate the spatially averaged turbulent momentum flux (i.e. the first term in Equation 1), a K-theory approach is employed:


$$\overline{\langle u' w' \rangle} = -K_m \frac{\partial \langle \overline{u} \rangle}{\partial z} \quad ; \quad \overline{\langle v' w' \rangle} = -K_m \frac{\partial \langle \overline{v} \rangle}{\partial z} \quad (2)$$

where $K_m$ is the diffusion coefficient for momentum using $k - l$ closure (Martilli et al., 2002) computed as:

$$K_m = C_k l_k \langle \bar{k} \rangle^{1/2} \qquad (3)$$

$C_k$ is a model constant for momentum, $l_k$ is a length scale, and $k$ represents the turbulent kinetic energy (TKE). A prognostic equation is employed to calculate the spatially averaged turbulent kinetic energy, assuming horizontal homogeneity as before.

$$\frac{\partial \rho \langle \bar{k} \rangle}{\partial t} = -\frac{\partial \rho \overline{\langle k' w' \rangle}}{\partial z} + \rho K_m \left[ \left( \frac{\partial \langle \bar{u} \rangle}{\partial z} \right)^2 + \left( \frac{\partial \langle \bar{v} \rangle}{\partial z} \right)^2 \right] - \rho \frac{g}{\theta_0} K_s \frac{\partial \langle \bar{\theta} \rangle}{\partial z} - \rho \varepsilon + \rho D_k \qquad (4)$$

where $D_k$ represents the source of $\langle k \rangle$ generated by the interaction between the buildings and the airflow, parametrized as $D_k = S(z) C_d |<\bar{u}>|^3$. Note that the buoyancy production term (third on the r.h.s.) is no longer zero as was the case of Santiago and Martilli, 2010. In this study, the diffusion coefficient is set to be 3.5 times higher than the momentum coefficient, consistent with the findings of Lu et al. (2024). The upper boundary condition for Eq. 4 is specified as a zero gradient. Buoyant effects directly influence $u$ and $v$ through source and sink terms within Eq. 4, which subsequently impact the turbulent diffusion coefficients in Eq. (3).

Dissipation is computed as:

$$\varepsilon = C_\varepsilon \frac{\langle k \rangle^{3/2}}{l_\varepsilon} \qquad (5)$$

$l_\varepsilon$ represents a length scale of dissipation and $C_\varepsilon$ is a model constant. According to Martilli et al., 2002, the values of the model constants $C_K$ and $C_\varepsilon$ are set to 0.4 and 0.71, respectively, based on the work of Bougeault and Lacarrere (1989). The two length scales in (3) and (5), $l_k$ and $l_\varepsilon$, are determined by solving the series of equations 9a to 9d reported in Martilli et al., 2002. Urban modifications of the length scales (i.e. both $l_k$ and $l_\varepsilon$) are applied as reported in equations 22 and 23 of the same study. Thus, the new length scale is added to the one computed using the traditional Bougeault and Lacarrere formulation, $l_{old}$:

$$\frac{1}{l} = \frac{1}{l_{old}} + \frac{1}{l_b} \qquad (6)$$

As in the formulation of Bougeault and Lacarrere, the turbulent coefficients for momentum and for heat are equal, in the proposed version of MLUCM BEP+BEM model, the turbulent coefficient for heat is estimated as: $K_s = K_m / P_{rt}$ according to Businger-Dyer relations' (Businger, 1988). The functional forms for $P_{rt}$ from Businger et al. (1971) and Dyer (1974) are:

$$P_{rt} = \begin{cases} \frac{0.74(1-9\zeta)^{-1/2}}{(1-15\zeta)^{-1/4}} & \zeta < 0 \\ \frac{0.74+4.7\zeta}{1+4.7\zeta} & \zeta > 0 \end{cases} \qquad (7)$$

where $\zeta = z/L$ is the stability parameter and Monin–Obukhov length ($L$) is estimated using the formulation of Louis (Louis, 1979). According to Högström (1988), who compared the Businger-Dyer relations with several other experimental studies and consistently used $k$=0.4 across data sets, $P_{rt}$ decreases as the atmosphere becomes more unstable. This decrease under unstable conditions is very robust (Li, 2019). To complete the 1-D MLUCM model, equations for vertical turbulent transport of spatial- and ensemble-average potential temperature and specific humidity are also solved. Under the assumptions of Eq. (1), the equation for conservation of potential temperature reduces to:

$$\frac{\partial \langle \bar{\theta} \rangle}{\partial t} = \frac{\partial}{\partial z}\left( K_s \frac{\partial \langle \bar{\theta} \rangle}{\partial z} \right) + S_{\theta R} + S_{\theta G} + S_{\theta Wl} + S_{\theta Wr} + S_{\theta B} \qquad (8)$$

the source terms $S_{\theta R}, S_{\theta G}, S_{\theta Wl}, S_{\theta Wr}, S_{\theta B}$ results from sensible heat exchange with roofs, the ground (canyon floor), walls (left and right sides of the canyon) and sensible heat generated by the cooling/heating system, respectively. The conservation equation for specific humidity is:

$$\frac{\partial \langle \bar{q} \rangle}{\partial t} = \frac{\partial}{\partial z}\left( K_s \frac{\partial \langle \bar{q} \rangle}{\partial z} \right) + S_{qR} + S_{qG} + S_{qV} + S_{qGR} + S_{qB} \qquad (9)$$

where $S_{qR}, S_{qG}, S_{qV}, S_{qGR}, S_{qB}$ are sources of moisture from built surfaces (i.e. roof, ground), canyon vegetation (both green area and street trees), green roof (where present) and latent heat generated by the cooling/heating system respectively. The flow of moisture from roofs and ground only occurs when water accumulates (in ponds) after rain events, as these surfaces are considered impermeable, while it is assumed that water cannot accumulate on vertical walls. MLUCM BEP+BEM is not coupled with an external land surface model. The latent heat flux results from the weighted average of contributions from natural (e.g., green road fractions and street trees.), and wet built surfaces (water from rain on road and roofs). For the green road fraction, the same scheme adopted by green roofs, described in Zonato et al. (2021), is used. For the trees, the latent heat flux is estimated using a simple empirical parameterization that partitions the radiation absorbed by leaves into sensible and latent heat. The upper boundary conditions for Eqs. 1, 8, and 9 are the time-varying horizontal wind component, potential air temperature, and specific humidity at the forcing height, respectively. The sensible heat fluxes from roofs ($S_{qR}$) and ground surfaces ($S_{qG}$) are calculated using the Louis (1979) approach based on Monin-Obukhov Similarity Theory, whereas the sensible heat fluxes from walls ($S_{\theta Wl}, S_{\theta Wr}$) are determined using a stability-independent bulk transfer method, which is dependent on wind speed (refer to Equations 15 and 16 in Martilli et al., 2002). In the proposed MLUCM BEP+BEM, the drag coefficient induced by buildings for mean wind speed and turbulent kinetic energy is estimated following the methodologies of Santiago and Martilli, 2010 as well as Gutiérrez et al., 2015. The drag coefficient is now modeled as a function of the building plan-area fraction as follows:

$$C_D(\lambda_p) = \begin{cases} 3.32\,\lambda_p^{0.47} & for\ \lambda_p \le 0.29 \\ 1.85 & for\ \lambda_p > 0.29 \end{cases} \qquad (10)$$

This approach provides a representation of the effect of buildings on air flow and turbulence, considering the variability of urban fabric and atmospheric stability. A methodology for determining the building plan-area fraction ($\lambda_p$) is described in Pappaccogli et al., 2021.

## 2.2 Green area and street trees

In this study, a module proposed by Stone et al. (2021) and Zonato et al. (2021) to represent street trees and green areas, respectively, has been integrated in MLUCM BEP+BEM.

## 2.2.1 Street canyon trees

The street tree canopy is modelled as a foliage layer within the urban canyon, positioned above street level. In the current configuration, street trees are not assigned a dedicated soil moisture reservoir, limiting the representation of tree-induced soil moisture dynamics and potentially introducing biases in the partitioning of turbulent heat fluxes. On the other hand, in most dense urban areas, the latent heat is a small component of the surface energy budget. The Beer-Bouguer-Lambert law is applied to account for radiation interception, which is based on the solar zenith angle. The amount of short-wave radiation reaching the street is computed as follows:

$$Rs_{street} = (1 - frac_{tree})Rs_{sun} + frac_{tree}Rs_{sun}e^{-\frac{0.5\sqrt{absvLAI}}{cosZr}} \qquad (11)$$

Where $Rs_{sun}$ is the radiation that would reach the street without trees, $frac_{tree}$ is the fraction of streets in the grid cell with trees, LAI is the Leaf Area Index, *absv* is the leaf absorptivity, and *Zr* is the solar zenith angle (Campbell and Norman, 2000). A similar relationship holds for the shortwave radiation reaching the walls below the tree crown. Radiation intercepted by the canopy is partitioned into sensible and latent heat production according to an empirical relationship provided by Michael Yonker (University of Illinois, Chicago - personal communication), as follows $\frac{Q_H}{Q_{le}} = 6.28 \cdot 10^{-4} \cdot SW_{down} - 9.643 \cdot 10^{-2}$, where $SW_{down}$ is the downward short wave radiation flux (W m$^{-2}$) at the top of the canopy. The report is based on observations from the AmeriFlux Sites US-MMS database. The height of the trees in the MLUCM BEP+BEM model follows the vertical level discretization and can be set by the user. The interaction of the canopy with radiation is limited to shortwave components, as a modelling simplification. In the current version, longwave interactions with the tree canopy are neglected. This includes the (computationally expensive) omission of longwave reflection and exchange between multi-layer 2D assemblages of buildings, roads, and tree foliage. Remarkably, the observed thermal radiation fluxes are accurately reproduced despite this simplification.

## 2.2.2 Street canyon gardens

The land surface scheme for street gardens is based on the land surface interaction parameterization used for green roofs in Zonato et al. (2021), adapted for street-level applications. Following the approaches of De Munck et al. (2013) and Gutierrez et al. (2015), it calculates energy and water budgets by considering factors such as incoming net radiation, water input from precipitation and irrigation, evapotranspiration from vegetation, heat exchange with the atmosphere, and energy and moisture diffusion throughout the soil. The model operates in one dimension, meaning that horizontal transport and subsurface flows are neglected. Each street garden is composed of ten distinct layers, with a total depth of 0.3 m. The upper five layers, collectively 0.08 m thick, represent the organic substrate where vegetation grows, allowing plant roots to extend to the bottom of this substrate. These plants are assumed to intercept all incoming radiation. Below the substrate is a 0.05 m thick drainage layer designed to remove excess water. The remaining four layers form the insulation layer, providing thermal protection for the system. Further details are provided in Zonato et al. (2021). The model simulates latent heat flux by accounting for both soil evaporation and transpiration through leaves, which absorb water from the substrate. Stomatal resistance depends on atmospheric conditions, water availability, and vegetation characteristics. Soil moisture transport is represented using the Richards' equation (Short et al., 1995), with moisture sources and sinks in the uppermost layer dependent on irrigation, precipitation, and evapotranspiration. No drainage is assumed at the garden's base.

Heat transfer between garden layers is calculated using the Fourier diffusion equation for soil temperature, with thermal diffusivity in natural roof layers dependent on soil moisture, similar to the green roof module. The modeled gardens are implemented with a simplified soil layer structure and a bottom boundary condition that does not account for infiltration into deeper permeable soil. While this setup may resemble planter boxes with impermeable bottoms, it does not aim to capture the full range of hydrological responses of gardens to precipitation and complex evapotranspiration processes.

A key improvement in this version of the MLUCM BEP+BEM model is the inclusion of vegetation, enabling more realistic urban scenarios. This enhances the model's ability to simulate urban vegetation dynamics, including green areas and street trees, and their impact on local microclimates. By accounting for vegetation-atmosphere interactions, the model improves the representation of local temperature and humidity patterns, supporting more comprehensive urban energy balance simulations. This is crucial for informing sustainable urban planning and climate adaptation strategies.

## 3 Methodology and simulation set-up

### 3.1 Measurement site

Simulations were conducted for the Preston area in Melbourne, Australia (AU-Preston; Lipson et al., 2022), which was also used in the PILPS-Urban Phase 2 project (Grimmond et al., 2011).

Observations at the AU-Preston site (Melbourne, Australia) were collected using sensors on a 40-meter eddy-covariance tower, which measured local-scale conditions. Data were recorded over 474.4 days (from 12 August 2003 to 28 November 2004),

with high-frequency measurements quality-controlled and averaged into 30-minute intervals. Quality control involved removing unsuitable periods and significant outliers (Lipson et al., 2024). In this study, the data are divided into two categories: forcing data (used to force the MLUCM BEP+BEM) and analysis data (used for model evaluation), as shown in Table 1. The analysis data are not gap-filled and are compared directly to observed values. In contrast, the forcing data are gap-filled using ERA5 global reanalysis data, with diurnal and seasonal adjustments applied to correct biases (Lipson et al., 2022). Since the MLUCM BEP+BEM model requires both direct and diffuse downward short-wave radiation as forcing data, the Spitters et al. (1986) method was used. This method estimates the direct fraction based on the solar angle and the ratio of measured to theoretical top-of-atmosphere radiation. Daytime flux errors at this site are estimated to be up to 10% (Best and Grimmond, 2015). Although extended evaluation periods help reduce random errors, systematic errors remain, as noted by Lipson et al. (2024).

| Variable | Description | Units | Positive |
|---|---|---|---|
| a. Forcing data | | | |
| $SW_{down}$ | Downward short-wave radiation | W m$^{-2}$ | Downward |
| $DSW_{down}$ | Direct downward short-wave radiation (Spitters et al., 1986) | W m$^{-2}$ | Downward |
| $FSW_{down}$ | Diffuse downward short-wave radiation | W m$^{-2}$ | Downward |
| $LW_{down}$ | Downward long-wave radiation | W m$^{-2}$ | Downward |
| $T_{air}$ | Air temperature | K | - |
| $Q_{air}$ | Specific humidity | kg kg$^{-1}$ | - |
| $P_{surf}$ | Station air pressure | Pa | - |
| Wind_N | Northward wind component | m·s$^{-1}$ | Northward |
| Wind_E | Eastward wind component | m·s$^{-1}$ | Eastward |
| Rainf | Rainfall rate | kg·m$^{-2}$·s$^{-1}$ | Downward |
| b. Analysis data | | | |
| $SW_{up}$ | Upward short-wave radiation | W m$^{-2}$ | Upward |
| $LW_{up}$ | Upward long-wave radiation | W m$^{-2}$ | Upward |
| $Q_{le}$ | Latent heat flux | W m$^{-2}$ | Upward |
| $Q_h$ | Sensible heat flux | W m$^{-2}$ | Upward |

| $Q_{tau}$ | Momentum flux | | N m$^{-2}$ | Downward |
|---|---|---|---|---|

**Table 2: Observational data description.**

## 3.2 Characterization of Preston site

The Preston area is primarily composed of single-family residential buildings with 1- or 2-storey heights, along with some
245 terraced commercial buildings of similar height. The vegetation includes a mix of trees and lawns (Lipson et al., 2024). The
neighbourhood falls within the Local Climate Zone 6 (LCZ6) classification, which represents open low-rise areas (Stewart and
Oke, 2012). The region's climate is temperate oceanic (CfB) according to the Köppen–Geiger system (Beck et al., 2018). The
values of the site parameters (Table 2) were sourced from various studies, as reported in Lipson et al. (2024), except for the
distribution of building heights. This parameter was calculated based on a digital surface model with a spatial and vertical
resolution of 1m (Figure 2a), according to (Lu et al., 2022).

| ID | Parameter | Value | Units | Footprint |
|---|---|---|---|---|
| | Baseline experiment parameters (1-7) | | | |
| 1 | Latitude | -37.73 | °N | Tower |
| 2 | Longitude | 145.01 | °E | Tower |
| 3 | Ground height | 93 | m | Tower |
| 4 | Measurement height above ground | 40 | m | Tower |
| 5 | Impervious area fraction | 0.62 | 1 | 500m radius |
| 6 | Tree area fraction | 0.225 | 1 | 500m radius |
| 7 | Grass area fraction | 0.15 | 1 | 500m radius |
| | Detailed experiment parameters (1-13) | | | |
| 8 | Roof area fraction | 0.445 | 1 | 500m radius |
| 9 | Building mean height | 6.4 | m | 500m radius |
| 10 | Tree mean height | 5.7 | m | 500m radius |
| 11 | Wall to plan area ratio | 0.4 | 1 | 500m radius |
| 12 | Resident population density | 2940 | Person km$^{-2}$ | Suburb average |
| 13 | Anthropogenic heat flux mean | 11 | W m$^{-2}$ | 500m radius |
| | Complex experiment parameters (1-14) | | | |

| 14 | Building distribution | - | 1 | 500m radius |
|---|---|---|---|---|

**Table 2: Site-descriptive metadata. Note: Parameters 1-7 were provided as inputs for "baseline" experiment, while "detailed" experiment allowed the use of all parameters (1-13) except the building distribution, which was used by "complex" experiment (1-14).**

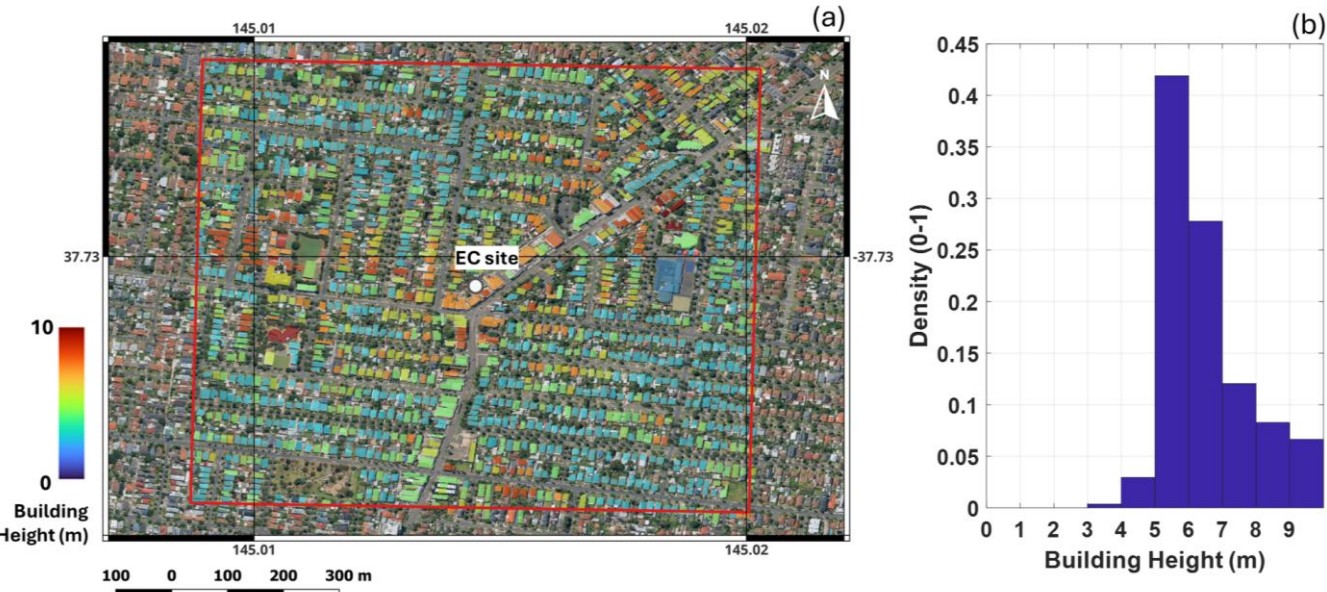

Figure 2: (a) Map of Preston site (Melbourne, Australia) with building heights, (b) fraction of buildings in a circular area with a radius of 500m around the observation as a function of their height (the 1m size of the bins corresponds to the vertical discretization of the model).

### 3.3 The MLUCM BEP+BEM configuration

In this study, the MLUCM BEP+BEM model was run offline with a vertical resolution of 1 meter up to 40 meters above ground level (AGL), using measurements (described in subsection 3.1) as the forcing data. The integration time step was set to 60 seconds, and the model computed meteorological variables from the top level of MLUCM BEP+BEM down to the ground (Fig. 1). Weather forcings were applied with the same integration time. All variables listed in Table 1 were provided from the observation site. To ensure equilibrium in soil states, a spin-up period of 10 years was performed, as outlined by Lipson et al. (2024). Urban material properties, including roof, wall and ground albedo, wall emissivity and roof and wall conductivity, were assigned according to the values given by Stewart et al. (2014) for class LCZ6. To ensure internal consistency across experiments and alignment with the Urban-PLUMBER protocol, a standardized set of LCZ6 parameters was adopted, enabling compatibility while acknowledging the need for future refinements to capture local variability.

The default window properties from the BEM model were adopted, with windows consisting of two 6-mm glass panels, each having an emissivity of 0.9 (Salamanca et al., 2010). To evaluate the impact of site-specific information on the model's performance, three experiments were conducted:

- **Baseline**: The model was set up using land cover and location data (Table 2, parameters 1-7) based on LCZ 6 morphological information. This experiment assessed models using data typically sourced from global high-resolution land cover datasets.

- **Detailed**: Parameters 1-13 in Table 2 were used for this experiment. This experiment tested whether performance improves with the inclusion of more difficult-to-obtain parameters (items 8-13 in the table) not widely available in general. The building distribution for "baseline" and "detailed" experiments were obtained by applying a gaussian distribution with mean values of 5m and 6.4m, respectively, and sigma of 3.

- **Complex**: All parameters listed in Table 2 were used to evaluate model performance improvements achieved by incorporating highly detailed urban geometry data, including the additional parameter 14 "Building distribution", which represents the fraction of buildings with roof heights within each vertical level (in this study the model vertical resolution is 1 metre, Figure 2b). For the "complex" experiment, the UT-GLOBUS database (Kamath et al., 2024) was employed to derive this parameter.

The validation of the MLUCM BEP+BEM model was conducted by comparing its outputs with "in situ" eddy covariance observational data from Preston, Melbourne (Australia) focusing on five surface energy fluxes: upward short-wave radiation ($SW_{up}$), upward long-wave radiation ($LW_{up}$), latent ($Q_{le}$) and sensible heat flux ($Q_h$), and momentum flux ($Q_{tau}$). All variables followed the ALMA protocol (Bowling and Polcher, 2001), consistent with prior PILPS projects, as outlined in Lipson et al. (2020). Validation results were submitted to the Model Evaluation Portal (https://modelevaluation.org) for comparison with observations (Abramowitz, 2018), including both automatic and manual diagnostics to identify potential configuration and output errors. Time series and Taylor diagrams were produced for the different experiments and models for an in-depth comparative analysis. As part of the Urban-PLUMBER Project (Lipson et al., 2024), key performance metrics such as bias, normalized mean error (NME), slope and correlation were evaluated on sub-hourly basis. The centered root-mean-square error (cRMSE), which accounts for variance and pattern discrepancies while excluding bias (Taylor, 2001), was also assessed. The performance of the MLUCM BEP+BEM model was also compared with one- and three-variable regression models, REG1 and REG3, which serve as out-of-sample empirical benchmarks. These benchmarks, based on statistical regressions using observational data independent of the test site, provide an unbiased evaluation of physical models across diverse climates and urban conditions.

Further comparisons were made with other similar parameterization schemes, including CM-BEM (Takane et al., 2022), TEB-SPARTCS (Schoetter et al., 2020, 2024), VTUF-3D (Lee and Park, 2008; Lee, 2011 and Lee et al., 2016) and BEPCOL (Martilli et al., 2002; Simón-Moral et al., 2017). Definitions of all performance metrics and additional details on benchmarks and models used in this comparison can be found in Appendix Table A1 and in "Participating model descriptions" Section of Lipson et al. (2024).

The model runs at one-minute time steps, with an average computational speed of approximately 4–5 ms per time step. A typical simulation covering one year of data requires approximately 30–40 minutes on a workstation using one Intel® Xeon®

Gold 5218 CPU @ 2.30GHz with 2 GB RAM, operating in a virtualized environment (VMware). A fully coupled mesoscale model (e.g., WRF with BEP+BEM; Vidal et al., 2021), typically requires more than one day for a 24 hours simulation using a single core (decreasing to 1 hour in a 64-core computer). Though computational costs may vary depending on the specific application and the hardware used, it is clear that MLUCM BEP+BEM offers an enormously reduced computational cost, enabling faster simulations and making it particularly suitable for long-term studies.

## 4 Results and discussion

### 4.1 Validation of surface-atmosphere fluxes

This section presents the evaluation of the MLUCM BEP+BEM performance in simulating key surface energy fluxes across the three experiments (i.e. "baseline", "detailed" and "complex"). By comparing modelled outputs with observed data, the accuracy and reliability of each experimental setup were assessed. Table 3 summarizes the performance metrics of the three

model experiments, benchmarked against other urban parameterization schemes (i.e. CM-BEM, TEB-SPARTCS, VTUF-3D and BEPCOL) and regression models (REG1 and REG3). Bold values highlight the experiments that achieved the best performance for the MLUCM BEP+BEM model, while underlined values indicate the model that outperformed all others overall. It is worth noting that the model parameterizations considered for comparison are based on "detailed" experiments and correspond to the latest submission in the ModelEvaluation.org application.

The evaluation of upward shortwave radiation ($SW_{up}$) reveals an improvement in model performance with increasing detail in urban parameterization. Among the three MLUCM BEP+BEM experiments, the "detailed" yields the best agreement with observations, exhibiting the lowest bias (1.51 W m$^{-2}$), normalized mean error (NME = 0.07), and a correlation of 1.00. The "baseline" experiment also performs well (bias = -1.68 W m$^{-2}$; SLOPE=1.02), followed by the "complex" experiment (Figure 3a), which shows slight overestimation (bias = 4.72 W m$^{-2}$). Overall, all the three experiments demonstrate an excellent ability

to represent this radiative component, although subtle improvements are observed for the "detailed" experiment. In comparison with other models, CM-BEM shows a comparative NME (0.10) with a slight positive bias, while BEPCOL achieves a similar correlation and NME and slightly lower bias (–3.02 W m$^{-2}$). The regression-based models REG1 and REG3 perform well in terms of bias but exhibit slightly lower slope values.

All three MLUCM BEP+BEM configurations perform similarly and robustly in simulating upward longwave radiation ($LW_{up}$),

with minimal differences in statistical indicators. The "baseline", "detailed", and "complex" experiments (Figure 3b) exhibit biases of 6.75 W m$^{-2}$, 9.68 W m$^{-2}$, and 7.24W m$^{-2}$, respectively, and an NME values ranged from 0.02 to 0.03. Correlation values are uniformly high (COR = 0.99), while slope values range from 0.84 ("complex") to 0.89 both for "baseline" and "detailed" experiments. These results suggest a strong model capability in capturing $LW_{up}$ across configurations, with the "baseline" experiment offering a slight advantage in all statistical measures analysed. Compared to external models, only

BEPCOL significantly overestimates $LW_{up}$ (bias = 15.92 W m$^{-2}$), while CM-BEM achieves the lowest bias (3.17 W m$^{-2}$) with

a slope above unity (1.23), indicating overfitting. Overall, MLUCM BEP+BEM outperforms other models in both accuracy and consistency for this radiative component.

The analysis of latent heat flux ($Q_{le}$) reveals that all three MLUCM BEP+BEM configurations slightly underestimate latent heat flux, with the "complex" experiment (Figure 3c) showing slightly improved agreement with observations in terms of correlation (0.64) and NME (0.88) compared to the "detailed" and "baseline" experiments. The SLOPE exhibits opposite trend, with values ranging from 0.54 ("detailed") to 0.64 ("baseline"). Compared to the regression models, the MLUCM BEP+BEM outperforms them across all statistical indices, except for correlation, which is slightly lower in all three experiments and BIAS, which is marginally higher in the "detailed" and "complex" experiments. Among the other models, CM-BEM and TEB-SPARTCS show similar BIAS, ranged from -3.05 W m$^{-2}$ to 4.82 W m$^{-2}$ respectively. In contrast, VTUF-3D and BEPCOL show larger underestimates than those of the MLUCM BEP+BEM model. Despite relying on generalized and standardized input data, the MLUCM BEP+BEM demonstrates a satisfactory ability to capture latent heat flux dynamics within the urban canopy layer, with overall performance broadly comparable to that of the other models although some limitations in reproducing higher flux magnitudes.

The three MLUCM BEP+BEM experiments display comparable performance in simulating sensible heat flux ($Q_h$), with all experiments slightly overestimating observed values. The "complex" experiment stands out with the lowest bias (17.86 W m$^{-2}$) (Figure 3d) and a slope of 1.03, indicating an accurate representation of flux magnitude. All experiments share a strong correlation coefficient (above 0.92), underlining the model's consistent ability to capture the temporal variability of $Q_h$ across different levels of urban detail. Compared to REG1 and REG3, the three MLUCM BEP+BEM experiments outperformed both regression models in many statistical indices. Among other models, TEB-SPARTCS achieves the lowest NME (0.35) and the highest correlation (0.94), whereas CM-BEM shows the lowest bias (6.48 W m$^{-2}$).

Further analysis indicates that the model reproduces observations more accurately during the daytime than at night (Fig. S1). The overall unsatisfactory performance of the model appears to be primarily due to the unrealistic simulation of nighttime fluxes, whereas daytime fluxes are reasonably well captured. As a result, the sensible heat flux is well reproduced when it represents a significant component of the surface energy budget, and less accurately reproduced at night, when its contribution is minimal. This discrepancy is not expected to significantly affect the estimation of quantities such as building energy consumption, which is in the focus of the model scope. Similar considerations apply to the latent heat flux, albeit to a lesser extent. These findings are consistent with the widely recognized limited role of latent heat fluxes in densely urbanized environments.

Regarding momentum flow ($Q_{tau}$), the three MLUCM BEP+BEM experiments provide consistent results in most statistical indicators, with negligible differences in bias (0.02–0.07 N m$^{-2}$), identical NME values (0.36), and consistent correlation coefficients (COR = 0.88), underlining the robustness of the model in representing this variable. The "complex" experiment (Figure 3e) shows a slightly improved slope (0.91), while the "detailed" and "baseline" experiments yield values of 0.78 and 0.87, respectively. Comparison with other models highlights the higher performance of the MLUCM BEP+BEM experiments, particularly regarding bias values, while correlation and slope are comparable to those of TEB-SPARTCS and BEPCOL. To

be noted, the regression models (REG1 and REG3) and the CM-BEM and VTUF-3D models do not provide data for this variable.

The MLUCM BEP+BEM model exhibits consistent and balanced performance across key components of the urban surface energy balance. Despite its reliance on standardized and relatively simplified input data, the model effectively reproduces both radiative and energy fluxes, with results broadly in line with those obtained from similar parameterizations schemes.

Both radiative components are accurately reproduced in all three experiments, highlighting the strong capability of the MLUCM BEP+BEM model to simulate them even when it is driven by standard input data. Turbulent fluxes, including latent and sensible heat, are characterized by some degree of slightly under- or overestimation, reflecting known challenges in urban modelling. Nonetheless, the MLUCM BEP+BEM model consistently captures their temporal dynamics, with statistical performance that remains competitive when compared to other similar models. Regarding momentum flux, all three MLUCM

BEP+BEM experiments confirm the model's ability to represent turbulence within the urban canopy, a crucial parameter for understanding wind behavior, heat transfer and pollutant dispersion in urban environments. Notably, the baseline configuration, based on LCZ-derived input, often performs comparably or better than more detailed experiments, highlighting the effectiveness of generalized classifications in representing key urban processes.

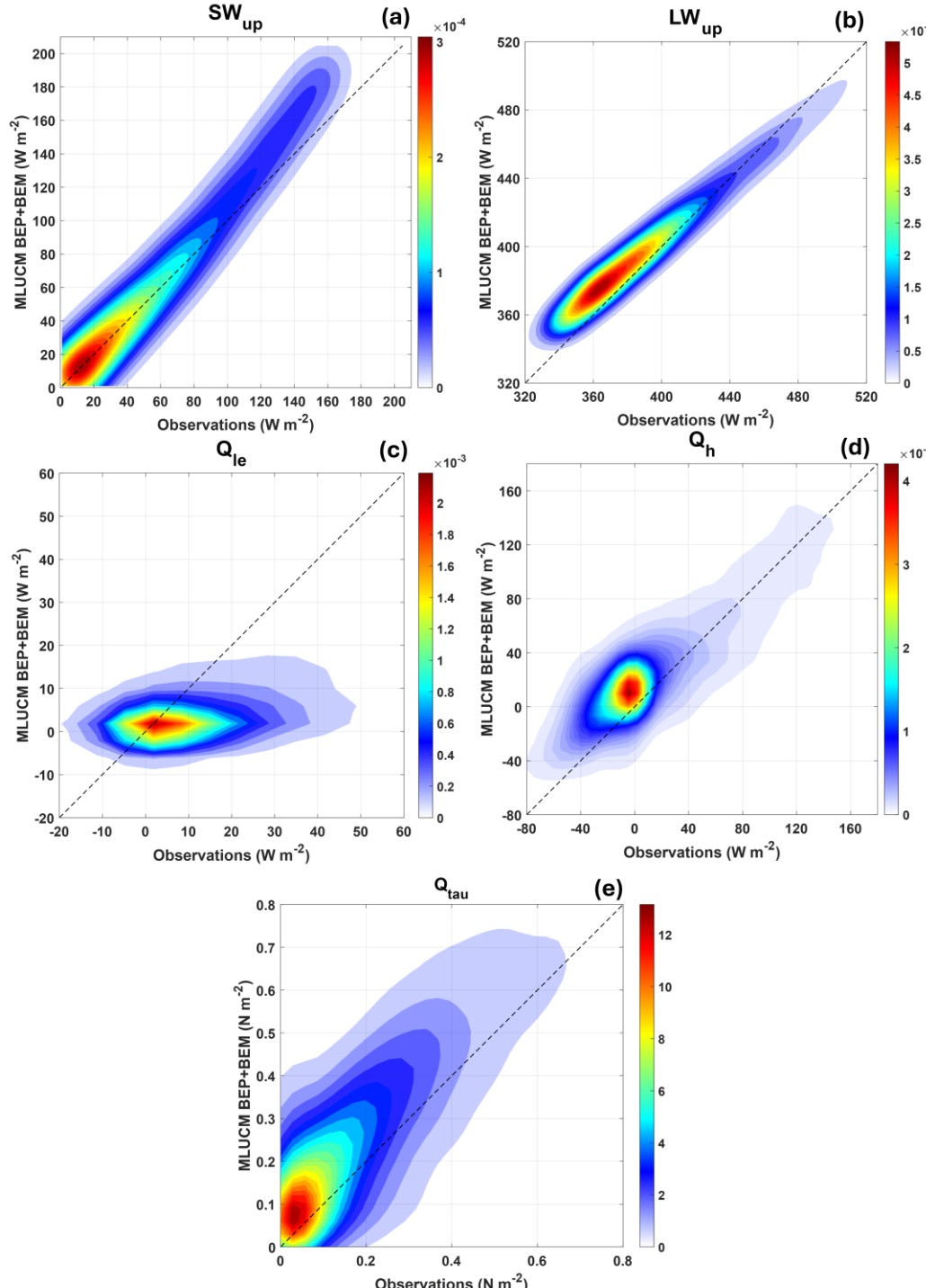

**Figure 3: Density scatter plot between observations and modelled MLUCM BEP+BEM "complex" experiment for (a) Upward short-wave radiation, (b) upward long-wave radiation, (c) latent heat flux, (d) sensible heat flux and (e) momentum flux. Density values represent the fraction of points per unit plot area (units $W^{-2}m^4$ for fluxes, $N^{-2}m^4$ for the momentum flux).**

| | | SW$_{up}$ | | | | | | | |
|---|---|---|---|---|---|---|---|---|---|
| | Baseline | Detailed | Complex | CM-BEM | TEB-SPARTCS | VTUF-3D | BEPCOL | REG1 | REG3 |
| BIAS | -1.68 | **1.51** | 4.72 | 2.83 | -8.77 | -15.90 | -3.02 | 0.31 | -0.98 |
| NME | 0.08 | **0.07** | 0.11 | 0.10 | 0.21 | 0.45 | 0.07 | 0.07 | 0.09 |
| SLOPE | **1.02** | 1.06 | 1.15 | 0.96 | 0.82 | 0.68 | 0.94 | 0.94 | 0.95 |
| COR | 0.99 | **1.00** | 0.99 | 0.98 | 0.99 | 1.00 | 1.00 | 1.00 | 0.99 |
| | | LW$_{up}$ | | | | | | | |
| BIAS | **6.75** | 9.68 | 7.24 | 3.17 | 7.79 | -4.82 | 15.92 | -1.58 | 11.29 |
| NME | **0.02** | 0.03 | **0.02** | 0.03 | 0.03 | 0.03 | 0.04 | 0.06 | 0.03 |
| SLOPE | **0.89** | **0.89** | 0.84 | 1.23 | 1.23 | 0.70 | 1.11 | 0.70 | 0.87 |
| COR | **0.99** | **0.99** | **0.99** | 0.96 | 0.98 | 0.96 | 0.97 | 0.75 | 0.98 |
| | | Q$_{le}$ | | | | | | | |
| BIAS | **-1.34** | -5.28 | -4.60 | -3.05 | 4.82 | -16.16 | -14.69 | -1.90 | -1.61 |
| NME | 0.89 | 0.94 | **0.88** | 0.77 | 0.62 | 1.69 | 1.25 | 0.78 | 0.79 |
| SLOPE | **0.64** | 0.54 | 0.56 | 0.55 | 0.65 | 0.19 | 0.32 | 0.35 | 0.34 |
| COR | 0.58 | 0.57 | **0.64** | 0.65 | 0.67 | 0.51 | 0.48 | 0.65 | 0.65 |
| | | Q$_H$ | | | | | | | |
| BIAS | 18.18 | 27.14 | **17.86** | 6.48 | 12.16 | -7.48 | 32.08 | 18.79 | 17.69 |
| NME | **0.47** | 0.49 | 0.48 | 0.64 | 0.35 | 0.52 | 0.52 | 0.54 | 0.53 |
| SLOPE | 1.15 | 1.16 | **1.03** | 0.66 | 1.14 | 0.73 | 1.08 | 0.81 | 0.81 |
| COR | 0.92 | **0.93** | 0.92 | 0.88 | 0.94 | 0.92 | 0.92 | 0.92 | 0.92 |
| | | Q$_{tau}$ | | | | | | | |
| BIAS | 0.06 | **0.02** | 0.07 | - | 0.52 | - | 0.43 | - | - |
| NME | **0.36** | **0.36** | **0.36** | - | 0.66 | - | 0.61 | - | - |
| SLOPE | 0.87 | 0.78 | **0.91** | - | 2.13 | - | 0.90 | - | - |
| COR | **0.88** | **0.88** | **0.88** | - | 0.88 | - | 0.82 | - | - |

**Table 3 Statistics of the three MLUCM BEP+BEM model experiments (i.e. baseline, detailed and complex), similar parameterisation schemes (i.e. CM-BEM, TEB-SPARTCS, VTUF-3D and BEPCOL) and benchmarks (i.e. REG1 and REG3) of the site metrics (NME: Normalized Mean Error, SLOPE: linear regression coefficient, COR: correlation)**

### 4.3 Taylor diagram evaluation

Taylor diagrams (Taylor, 2001) provide a visual summary of the agreement between a set of patterns and observations. Similarity is evaluated using three key metrics: correlation, which measures the strength of the relationship; centered root-mean-square error (cRMSE), which assesses the magnitude of discrepancies; and standard deviation, which reflects the amplitude of variations. These diagrams are particularly useful for evaluating different aspects of complex models and comparing the performance of multiple models (e.g. IPCC, 2001).

Starting with SW$_{up}$ (Figure 4a), all models show strong agreement with observations, with correlation coefficients (R) above 0.99 (except CM-BEM) and cRMSE values below 15 W m$^{-2}$. The MLUCM BEP+BEM model performs well, matching the

observed $SW_{up}$ amplitude, with a standard deviation close to the observed data, especially for the "baseline" experiment. Notably, the three MLUCM BEP+BEM experiments exhibit a low cRMSE, even when compared to the other models, except for BEPCOL, which achieves the highest accuracy with a cRMSE of 3.91 W m⁻². However, despite the overall good fit, all models slightly underestimate the observed amplitude, except for MLUCM BEP+BEM, which slightly overestimates it.

For $LW_{up}$ (Figure 4b), all the models show excellent agreement with measurements, exhibiting a cRMSE values consistently below 20 W m⁻². The MLUCM BEP+BEM model accurately captures the $LW_{up}$ amplitude, with standard deviations closely matching observations across different experiments, ranged from 35.58 W m⁻² for "complex" to 37.70 W m⁻² for "detailed". Notably, the three experiments produce consistent results, indicating that the model's complexity does not significantly impact its ability to simulate $LW_{up}$ variations. Among the other models, all exceed a cRMSE of 10 W m⁻², with a higher standard deviation than observed, except for the VTUF-3d model, which underestimates it.

Regarding $Q_{le}$ (Figure 4c), all models struggle to accurately represent this variable, with all the models underestimating the amplitude, except for the "baseline" experiment. The MLUCM BEP+BEM model effectively captures the variability of latent heat flux, with STDs values ranging from 57.19 W m⁻² in the "baseline" experiment to 45.69 W m⁻² in the "complex" configuration. Conversely, the cRMSE is lowest for the "complex" case. Compared to the other models, the three MLUCM BEP+BEM experiments achieve comparable cRMSE and STDs values to CM-BEM and TEB-SPARTCS. In terms of correlation, the "complex" experiment performs similarly to the TEB-SPARTCS and VTUF-3D models, while the other experiments show larger errors, though still within a consistent range.

For sensible heat flux ($Q_h$, Figure 4d), the MLUCM BEP+BEM model shows good overall agreement with observations, with standard deviation values ranging from 103.44 W m⁻² in the "complex" experiment to 115.10 W m⁻² in the "baseline." Among all experiments, the "complex" configuration performs best, yielding the lowest STD and cRMSE values. Similarly, models such as TEB-SPARTCS and BEPCOL effectively capture the observed variability, whereas CM-BEM and VTUF-3D tend to underestimate the flux amplitude.

For momentum flux ($Q_{tau}$, Figure 4e), all three MLUCM BEP+BEM experiments closely match the observed variability, with standard deviations around 0.29–0.34 N m⁻², in line with the observed value (0.33 N m⁻²). Correlation coefficients remain consistently high (0.88), and cRMSE values are low (0.16-0.17), indicating strong agreement across configurations. Among external models, BEPCOL performs reasonably well, though with slightly lower correlation (0.82) and a higher cRMSE (0.21). In contrast, VTUF-3D overestimates variability (STD = 0.80) and exhibits reduced accuracy (cRMSE = 0.53). CM-BEM and TEB-SPARTCS do not provide output for this variable. MLUCM BEP+BEM demonstrates robust skill in representing momentum exchange within the urban canopy layer.

Overall, Taylor's analysis shows that the MLUCM BEP+BEM model performs reliably across all components of the surface energy balance, remaining competitive with comparable models. For both shortwave ($SW_{up}$) and longwave ($LW_{up}$) radiation, the model consistently aligns with observations across all experiments. In the case of latent heat flux ($Q_{le}$), although amplitude remains difficult to capture for all models, the "complex" setup of MLUCM BEP+BEM shows improved accuracy in representing its variability. Sensible heat flux ($Q_h$) is also well reproduced, though slightly overestimated, with the "complex"

configuration showing the best agreement with observed variability and error metrics, reflecting a consistent improvement from the "baseline" to the "detailed" and finally the "complex" experiment. Momentum flux ($Q_{tau}$) is accurately represented in all configurations, with the model closely matching observed values and outperforming most other similar models.



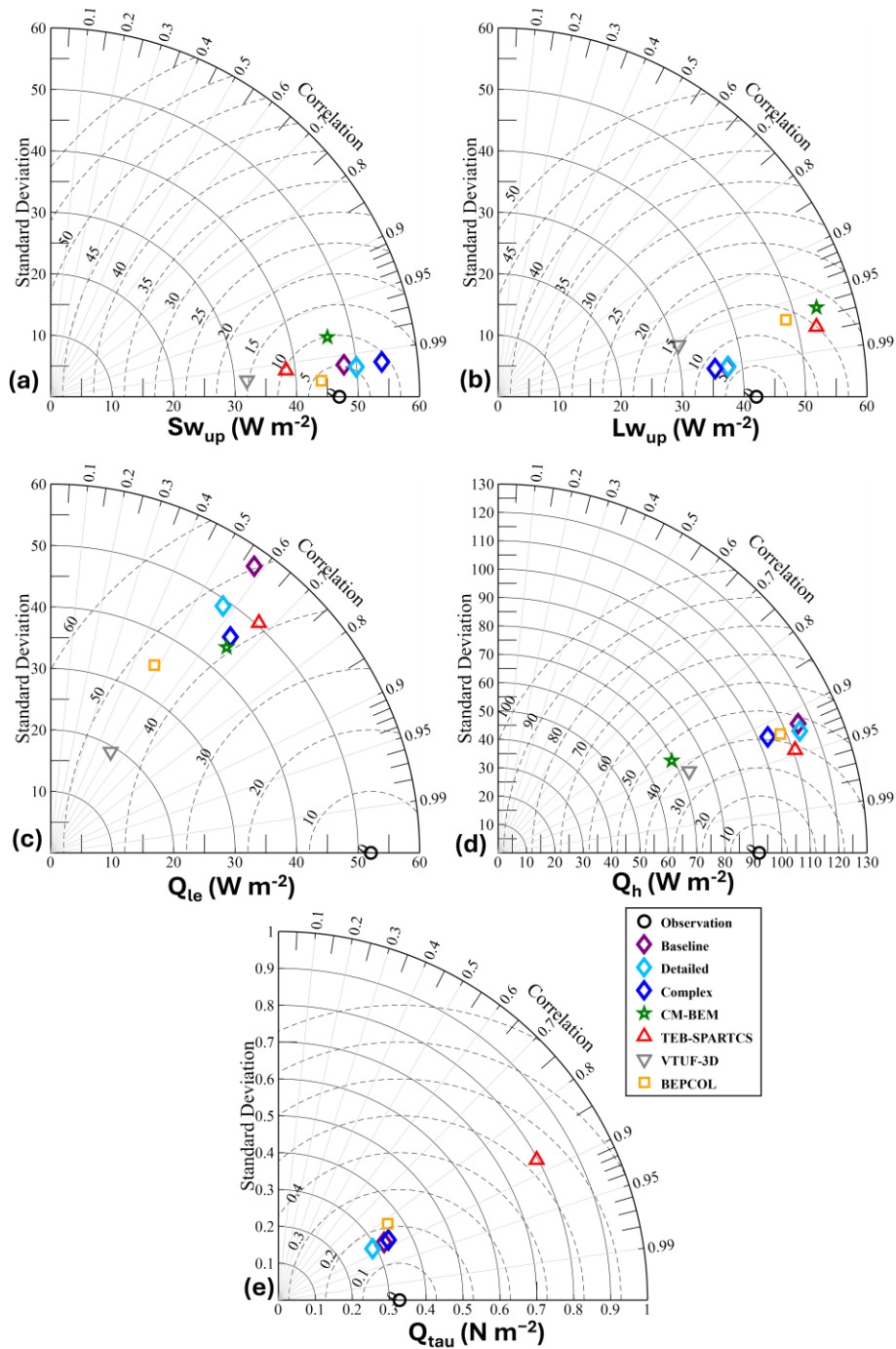

**Figure 4** Taylor diagrams for (a) Upward short-wave radiation, (b) upward long-wave radiation, (c) latent heat flux, (d) sensible heat flux and (e) momentum flux, based on model results at 30-minute intervals. The Taylor diagram represents standard deviation (σ), correlation, and cRMSE (dashed circles). The distance between model results (depicted with coloured markers) and observations (depicted with the black circle at the base of the diagram) represents their RMSE and the angular coordinate their correlation. Models with large accuracy are located near to the observations.

**5 Discussion and conclusions**

The development of the MLUCM BEP+BEM model represents a significant advancement in urban microclimate modelling. It integrates the vertical turbulent diffusion scheme of Santiago and Martilli (2010) with the Building Effect Parameterization (BEP; Martilli et al., 2002) and the Building Energy Model (BEM; Salamanca et al., 2010). Moreover, it incorporates enhancements in turbulent length scales for dissipation and eddy coefficients, accounting for atmospheric stability inspired by the turbulence scheme of Bougeault and Lacarrere (1989) turbulence scheme. Finally, the model includes green areas and

street trees following the methods of Zonato et al. (2021) and Stone et al. (2021), respectively.

Validation of the MLUCM BEP+BEM model was conducted using data from the Urban-PLUMBER project, demonstrating its capability to reproduce surface-atmosphere fluxes at a suburban site in Preston, Melbourne, Australia. To understand the impact on the sensitivity of the model, three experiments ("basic", "detailed" and "complex") fed with different levels of detail of urban parameters were evaluated. The model exhibits strong agreement with observed data, particularly in simulating

shortwave ($SW_{up}$) and longwave ($LW_{up}$) upward radiation, with high correlations and low errors across different model configurations.

Energy fluxes are well captured, although some under- and over-estimates are observed for latent ($Q_{le}$) and sensible ($Q_h$) heat fluxes respectively, while consistent results for all three experiments were observed for momentum flux ($Q_{tau}$). The "baseline" experiment shows a mixed behavior in reproducing sensible heat fluxes ($Q_h$), with reduced BIAS but overestimated variability

compared to the more detailed experiments. This is mainly attributed to its representation of a compact and low urban structure derived from the LCZ classification, which reduces heat accumulation and release dynamics.

The discrepancies observed in both sensible ($Q_h$) and latent heat flux ($Q_{le}$) is probably attributable to the limitations of the current model setup, which omits the presence of trees in green areas. As a result, the model does not account for the full extent of vegetative cover, particularly in areas with dense greenery. In addition, street trees may receive limited solar radiation due

to shading effects and potential inaccuracies in urban geometry representation, which can further reduce their transpiration capacity. The absence of a dedicated soil moisture reservoir for trees also limits the simulation of their evapotranspirative contribution. These factors can contribute to an imbalance in the surface energy budget. The absence of observational data for ground heat flux ($Q_g$) prevents direct validation, limiting the evaluation of model performance for this component and warranting caution in interpreting its impact in coupled simulations. The heat fluxes inaccuracy is expected to be not important

during daytime when heat fluxes are large for estimating quantity such as energy building consumption, which is in the focus of the model scope.

Overall, the MLUCM BEP+BEM model exhibits a strong capability in simulating the urban energy balance, effectively accounting for the influence of buildings, urban structures and vegetation. Notably, when provided with varying urban parameters, the model excels at representing turbulent momentum flux, a key factor in understanding wind pattern, heat

transfer, and pollutant dispersion in urban environments.

Results show that the integration of detailed, site-specific information on urban elements such as building geometry and vegetation lead to some improvements in terms of correlation, cRMSE, and STD in the simulation of latent and sensible heat

fluxes. This level of detail is particularly important for applications involving building energy demand or urban mitigation strategies, where accurate representation of urban morphology, especially the volumetric structure, is essential (Pappaccogli et al., 2021).

These findings align with those of Lipson et al., (2024), which indicate that more complex models benefit from comprehensive data inputs to more accurately describe the surface energy budget.

MLUCM BEP+BEM is driven by the atmospheric variables (i.e. Downward direct-diffuse short-wave radiation, downward long wave radiation, air temperature, specific humidity, air pressure, Northward-Eastward wind components and rainfall rate) at its top boundary and can be either coupled with meteorological and climate models or be operated offline.

Although the current model adopts simplified representations of vegetation and hydrological processes, the results demonstrate good agreement with, and in some cases outperform, other comparable urban parameterizations. These findings suggest that the model can provide a reasonable representation of key urban climate dynamics, even when driven by limited input detail. Its computational efficiency makes it particularly suited for exploring long-term trends and assessing mitigation strategies focused on the thermal and radiative properties of the built environment, such as the implementation of green and cool roofs, photovoltaic panels, or energy retrofitting measures. Conversely, care should be taken when using the model to evaluate the role of gardens and street trees, due to the simplified treatment of soil hydrology. In such cases, more detailed models such as the Urban Tethys-Chloris (UT&C) model (Meili et al., 2020), which explicitly account for ecohydrological processes, may provide a more accurate representation of vegetation effects on urban climate. However, MLUCM BEP+BEM represents a promising tool for supporting urban-scale climate analyses and informing decision-making processes. Further research includes experiments forcing the MLUCM BEP+BEM model with the ERA5 reanalysis to assess its sensitivity to various input parameters, including urban morphology and vegetation characteristics. Moreover, the model will be forced with climate projections to investigate the impact of climate change on the different urban processes, such as overheating, building energy demands, outdoor thermal comfort, and the efficacy of adaptation strategies, including urban greening, green and cool roofs, photovoltaic panels and hybrid sustainable infrastructure.

**Code/Data availability**

The code of MLUCM BEP+BEM can be accessed at https://doi.org/10.5281/zenodo.14773142 (Pappaccogli, 2025a). The record is publicly accessible. A user manual with key information on how to use MLUCM BEP+BEM is available in the same repository.

The results of the simulation over Preston, Melbourne (Australia) shown in the paper are stored at https://doi.org/10.5281/zenodo.15731694 (Pappaccogli, 2025b).

## Author contributions

PL initial definition of research aims; GP, AZ and AM designed the methodology and developed the model code; GP developed the MLUCM BEP+BEM code with the support of AZ and AM; GP and AZ performed the MLUCM BEP+BEM simulations; GP prepared the figures and the paper with contributions from all co-authors; GP, AZ, AM, RB and PL analyzed the data, reviewed and edited the manuscript.

## Competing interests

The authors declare that they have no conflict of interest.

## Acknowledgements

This work was carried out within the framework of the ICSC – Centro Nazionale di Ricerca in High Performance Computing, Big Data and Quantum Computing, funded by the European Union – NextGenerationEU (CUP F83C22000740001). Acknowledgments are extended to Mathew Lipson for his contribution to the preparation of the data analysis for the ModelEvaluation.org application and for conducting a detailed assessment of the results, which significantly enhanced the accuracy of the proposed model. Appreciation is also extended to Jiachen Lu for providing the modelled site geometry dataset, and to IT technician Luigi Marzo for his support in managing the computational infrastructure.

## Financial support

This work was supported by ICSC – Centro Nazionale di Ricerca in High Performance Computing, Big Data and Quantum Computing, funded by European Union – NextGenerationEU (CUP F83C22000740001).

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
