# Peer review of "MLUCM BEP+BEM: An offline one-dimensional Multi-Layer Urban Canopy Model based on the BEP+BEM scheme"

_EGUsphere, 2025_

## Referee Comment (RC1)

Review of the manuscript entitled "MLUCM BEP+BEM: An offline one-dimensional Multi-Layer Urban Canopy Model based on the BEP+BEM scheme" by G. Pappaccogli et al. This manuscript describes and evaluates the performance of the most recent version of the one-dimensional multilayer urban canopy model (MLUCM) BEP+BEM. This upgraded version incorporates new features such as gardens and trees in street canyons, making it an ideal tool for evaluating adaptation and mitigation strategies to combat global warming. The BEP+BEM model is evaluated offline by comparing simulated against observed turbulent fluxes collected during an intensive observational period from August 2003 to November 2004 in a suburb of Melbourne (Australia). The model is forced with downwelling shortwave and longwave radiation, air temperature, specific humidity, wind speed, pressure, and precipitation, and its performance is evaluated by comparing modeled upwelling shortwave and longwave radiation, sensible and latent heat, and momentum flux against observations.

Three different urban scenarios are investigated with the BEP+BEM scheme based on the degree of site-specific urban morphology data used in the simulations. The first experiment (baseline simulation) uses site-specific observations for the impervious area fraction, tree area fraction, and grass area fraction. In this numerical experiment, the prescribed values for the LCZ6 (Local Climate Zone 6) urban category are used for the rest of urban parameters. The second experiment (detailed simulation) utilizes site-specific observations for six additional urban parameters. These parameters are the roof area fraction, mean building height, mean tree height, wall to plan area ratio, population density, and anthropogenic heat flux. Finally, the third experiment (complex simulation) is identical to the previous detailed simulation except for the building height distribution that is characterized with site-specific observations instead of prescribed values from the LCZ6 classification.

Results demonstrate (based on table 3 of the manuscript) that the complex simulation produced the best correspondence against observations for the upwelling shortwave and longwave radiation, the baseline experiment for the sensible and latent heat flux, and the detailed experiment for the momentum flux. Overall, the performance of the three numerical experiments was excellent (except for the latent heat flux that was considerably underestimated) and the differences among them were not significant. This article is interesting and represents an advance in urban climate modeling, but some clarifications are needed before it can be accepted for publication:

1) It is not clear if the MLUCM BEP+BEM is coupled to a land surface model to simulate land-atmosphere interactions for natural/rural surfaces. What are the sources of latent heat in BEP+BEM? How is the grid-averaged latent heat flux estimated?

2) Have the street-canyon-trees and street-canyon-gardens models been validated in a previous work? Are trees and gardens (in the street canyon) the unique sources of latent heat in these simulations?

3) Table 1 shows that tree area fraction is 0.225 for this neighborhood. How do sensible and latent heat fluxes change when this fraction is set to 0? In other words, is the role of trees important for mesoscale simulations? Was the impact of trees on near-surface air temperature and/or specific humidity modeled in the numerical experiments?

4) Similarly, table 1 shows that grass area fraction is 0.15 for this neighborhood. How do sensible and latent heat fluxes change when this area fraction is set to 0? Was the impact of grass area fraction on near-surface air temperature and/or specific humidity modeled in the simulations?

5) Overall, the baseline experiment produced good results compared to the other two experiments, which could indicate that some urban parameters are more important than others to accurately simulate the urban climate. Could you explain what is the added value of considering site-specific observations for six/seven additional urban parameters (compared to the baseline experiment) for mesoscale climate simulations?

---

## Referee Comment (RC3)

Review - MLUCM BEP+BEM: An offline one-dimensional Multi-Layer Urban Canopy Model based on the BEP+BEM scheme

**Summary:** This manuscript, entitled MLUCM BEP+BEM: An offline one-dimensional Multi-Layer Urban Canopy Model based on the BEP+BEM scheme" by G. Pappaccogli et al., describes the performance of a one dimensional multilayer urban canopy model (MLUCM) BEP+BEM. The comparison uses a standard dataset used for benchmarking from Melbourne Australia, and the model is forced by common weather and climate outputs (downwelling shortwave and longwave radiation, air temperature, specific humidity, wind speed, pressure, and precipitation). The model outputs of upwelling shortwave and longwave radiation, sensible and latent heat, and momentum flux are compared against observations. Three separate changes are made to show off MLUCM. The first uses site-specific observations of important urban fractions and prescribed values from local climate zone 6 (baseline). A second experiment adds in 6 more parameters, while the last simulation uses an additional building height distribution factor on top of the extra parameters.

Results show that this model is good at representing upwelling shortwave and longwave radiation, as well as the momentum flux due to the highly technical treatment of these factors. The performance of the other fluxes, sensible and latent, are lacking in the same skill, and ground heat flux was not mentioned at all. I agree that this article is worth publishing, but there are clarifications and extra work needed to ensure that a more complete picture is given to potential users of this code base. Specific attention should be paid to the "why" behind insensitivity to the sensible heat fluxes to different parameters and the lack of skill in the latent heat flux values. Specifically, I suggest:

**Major points (in order of text, not in order of importance):**

1. Is it a good assumption to not let longwave radiation interact with the tree canopy processes? Longwave radiation should be absorbed, transmitted, and emitted from the tree canopy like all other structures, which would then affect the radiative temperature and therefore heat flux partitioning. Please clarify what you mean by "The canopy interacts only with short-wave radiation and does not affect long-wave radiation components." on line 187.

2. Why use an empirical formulation for the partitioning of heat fluxes dependent on the shortwave radiation? Do street trees not have their own soil moisture stores that are similar to the street canyon gardens? Not taking into account the changes in soil moisture induced by urban trees, and the reduction in latent heat and subsequent

changes to sensible and ground heat fluxes, could be biasing the results of this study (e.g. a reason why there are such large discrepancies in the latent heat results).

3. Do you believe that the LCZ6 parameters you chose are representative for this space? Did you check the albedo and emissivity against remotely sensed averages? To my knowledge, LCZ give a range of values to select from, but these are likely to change given the age and type of architecture chosen for the study region.

4. Upon reading the street canyon gardens section, I think that clarification is needed to discuss whether street trees are treated similarly (e.g. using ecohydrologic principals) or not.

5. I think that a more detailed investigation of the "why" behind sensible and latent differences in the single layer BEP BEM model is needed. For instance, why is the baseline doing better in the sensible heat flux compared to the more detailed versions of the model presented? Is there a lack of sensitivity/too much sensitivity to the parameters that were introduced? For the latent heat flux, it is tricky to tell what is going on without a better explanation of how green areas are represented. Is there soil moisture/ hydrology being simulated? Or is this the ratio that was mentioned in the methods section? Examining the code shows that partitioning between sensible and latent heat fluxes for trees, which would be the major contributor to the latent heat signal, is using this ratio. More justification is needed on why this is appropriate given the biases that it introduces, given that urban trees do increase latent heat fluxes to be higher than those shown (even in modeling experiments, like related work with BEP-Tree (https://doi.org/10.1016/j.uclim.2020.100590) or tiling approaches in Noah-MP HUE that represent ecohydrology (**https://doi.org/10.1029/2023WR035511**)).

6. Authors could give a more detailed breakdown of what they hypothesize is going wrong than "whose cause deserves further investigations, are present for latent heat flux (Qle)." As stated on line 414.

7. What is going on with the ground heat flux? I am assuming there are no measurements, but do the results from this flux look believable? I would think that because of the errors in Qh and Qle, Qg would be also too high, and thus could cause a warming feedback when introduced into weather or climate models like suggested in the discussion.

8. The final point of this paper, "Further research includes experiments forcing the MLUCM BEP+BEM model with the ERA5 reanalysis to assess its sensitivity to various input parameters, including urban morphology and vegetation characteristics. Moreover, the model will be forced with climate projections to investigate the impact of climate change on the different urban processes, such as overheating,

building energy demands, outdoor thermal comfort, and the efficacy of adaptation strategies, including urban greening, green and cool roofs, photovoltaic panels and hybrid sustainable infrastructure." is a lofty goal, and I would agree that this model is able to look at urban morphology pretty neatly. The issue is coming from the vegetation characteristics, urban greening, green and cool roofs, and the interactions with urban comfort and other applications. As of right now, the latent heat flux and sensible heat fluxes are wrong, which would then cause these to be erroneous. This model is on the right track, but there needs to be more justification/investigation/discussion on how we could use this model to look at urban climate adaptation with the errors that are present within the model right now.

9. When investigating the model code, the code is clean but there are some missed opportunities to give an indication of what each of the subroutines are doing. Consider adding those so that folks who want to add/modify this code base will know what is going on in each routine and call to the routine!

**Minor points:**

1. Line 59: missing a space between "1 Dimensional"

---

## Author Comment (AC5)

**Reply on RC1**

Replies to the Reviewers' remarks

We thank all the Reviewers for the careful revisions and the constructive comments to our work. All suggestions have been appreciated and have been taken into careful consideration in our revision. Below, we provide a detailed reply to all points raised by the reviewers.

**Referee: 1**

1) It is not clear if the MLUCM BEP+BEM is coupled to a land surface model to simulate land-atmosphere interactions for natural/rural surfaces. What are the sources of latent heat in BEP+BEM? How is the grid-averaged latent heat flux estimated?

**R**: In order to clarify this point the following sentence has been added to the manuscript in line 157-162:
*"MLUCM BEP+BEM is not coupled with an external land surface model. The latent heat flux results from the weighted average of contributions from natural (e.g., green road fractions and street trees.), and wet built surfaces (water from rain on road and roofs). For the green road fraction, the same scheme adopted by green roofs, described in Zonato et al. (2021), is used. For the trees, the latent heat flux is estimated using a simple empirical parameterization that partitions the radiation absorbed by leaves into sensible and latent heat."*

2) Have the street-canyon-trees and street-canyon-gardens models been validated in a previous work? Are trees and gardens (in the street canyon) the unique sources of latent heat in these simulations?

**R**: The street-canyon-trees and street-canyon-gardens models, although previously proposed and applied in the studies of Stone et al., (2021) and Zonato et al., (2021), respectively, had not been validated with respect to the latent heat flux. While the street-canyon gardens model is inspired by the parameterization proposed by De Munck et al. (2013), which was validated in their original study, the street-canyon trees model has not undergone specific validation until now. The results presented in this study are the first direct validation against in situ observations of latent heat flux and show a quality comparable to other models.
Street-canyon-trees and gardens are the main latent heat source in the three experiments analyzed in this study, but these simulations also consider the contribution from artificial wet surfaces. Further our model can include other sources, such as air conditioning systems and green roofs (not active in the simulations presented in this paper), as well as evaporation of rain water accumulated on roofs and roads.

de Munck, C. S., Lemonsu, A., Bouzouidja, R., Masson, V., and Claverie, R.: The GREENROOF module (v7.3) for modelling green roof hydrological and energetic performances within TEB, Geosci. Model Dev., 6, 1941–1960, https://doi.org/10.5194/gmd-6-1941-2013, 2013.

3) Table 1 shows that tree area fraction is 0.225 for this neighborhood. How do sensible and latent heat fluxes change when this fraction is set to 0? In other words, is the role of trees important for mesoscale simulations? Was the impact of trees on near-surface air temperature and/or specific humidity modeled in the numerical experiments?

R: Simulations were conducted for the "complex" experiment, considering three scenarios: one including only street trees (referred to as **TREES**), one including only gardens (**GARDENS**), and one excluding both types of vegetation (**NOVEG**). These configurations were designed to address this and the following research question. Additionally, we identified an error in the previously used tree area fraction value of 0.225. This value should have been weighted by the street fraction according to the model's infrastructure settings. For further clarification, please refer to the general comment, also provided in response to Reviewer 2. The interpretation of the changes of variability in the Taylor results indicates that excluding trees significantly reduces the latent heat flux and at the same time it decreases the reflected shortwave flux. Concerning the sensible heat flux the latter effect prevails determining its increase.

Near-surface air temperature and humidity measurements are not included in the Urban-PLUMBER protocol, making it difficult to establish a reliable reference for validating model output and answering the last question of the Reviewer.

[Figure]

|  | Complex | NOVEG | TREES | GARDENS |
|---|---|---|---|---|
| $Q_{le}$ | | | | |
| BIAS | -4.60 | -28.62 | -0.69 | -22.49 |
| NME | 0.88 | 5.11 | 0.86 | 2.58 |
| SLOPE | 0.56 | 0.05 | 0.66 | 0.19 |
| COR | 0.64 | 0.22 | 0.62 | 0.65 |
| $Q_h$ | | | | |
| BIAS | 17.86 | 21.76 | 26.6 | 6.28 |
| NME | 0.48 | 0.47 | 0.48 | 0.64 |
| SLOPE | 1.03 | 1.17 | 1.39 | 0.67 |
| COR | 0.92 | 0.92 | 0.92 | 0.89 |

4) Similarly, table 1 shows that grass area fraction is 0.15 for this neighborhood. How do sensible and latent heat fluxes change when this area fraction is set to 0? Was the impact of grass area fraction on near-surface air temperature and/or specific humidity modeled in the simulations?

R: The interpretation of the changes in variability in the "GARDENS" simulation in the Taylor results (olive stars) indicates that excluding gardens reduces the latent heat flux (but to a lower extent than excluding trees) and has a small effect (increase) on the reflected shortwave flux. The sensible heat flux decreases to compensate for the increased latent heat.

5) Overall, the baseline experiment produced good results compared to the other two experiments, which could indicate that some urban parameters are more important than others to accurately simulate the urban climate. Could you explain what is the added value of considering site-specific observations for six/seven additional urban parameters (compared to the baseline experiment) for mesoscale climate simulations?

R: The results indicate that "detailed" and "complex" experiments, incorporating more detailed morphological parameters, lead to better estimates of sensible and latent heat flux, often exhibiting smaller errors than the "baseline" experiment.
However, the added value of incorporating detailed urban descriptors would become evident when addressing aspects such as building energy demand or the effectiveness of urban mitigation strategies (which are not considered in this study that refers to the Urban-PLUMBER protocol). As emphasized in previous work (Pappaccogli et al., 2021), accurately representing urban morphology, particularly the volumetric structure of the built environment and the exchange surfaces between built-up areas and the urban boundary layer is crucial for simulating energy demand. This is directly linked to anthropogenic heat emissions, which are a key component of urban climate dynamics.

Additional clarification has been included in the Section 5 of the manuscript in lines 460-463 as follows:

"*Results show that the integration of detailed, site-specific information on urban elements such as building geometry and vegetation generally improves the simulation of energy fluxes. This level of detail is particularly important for applications involving building energy demand or urban mitigation strategies, where accurate representation of urban morphology, especially the volumetric structure, is essential (Pappaccogli et al., 2021).*"

Pappaccogli, G., Giovannini, L., Zardi, D., & Martilli, A. (2021). Assessing the ability of WRF-BEP + BEM in reproducing the wintertime building energy consumption of an Italian Alpine city. Journal of Geophysical Research: Atmospheres, 126, e2020JD033652. https://doi.org/10.1029/2020JD033652

General comments for reviewer 1

The Reviewers' comments offered an opportunity to conduct a more in-depth investigation into the representation of latent heat flux in our model. This led to the discovery of a significant error: the tree and garden area fractions used in the three experiments were only 55% and 76%, respectively, of the correct values. This discrepancy systematically underestimated both the magnitude and variability of the latent heat flux.

Furthermore, the recently proposed UT-GLOBUS database by Kamath et al. (2024) was employed to enhance the representation of building distributions in the "complex" experimental setup, producing a generalized increase of the height of buildings. These two changes have led to a reduction of the negative latent heat bias of our former simulations. The updated simulations yield results that are consistent with those obtained from similar parameterization schemes.

Additional clarification has been included in the manuscript in lines 277-278 as follows:
"*For the "complex"* experiment*, the UT-GLOBUS database (Kamath et al., 2024) was employed to derive this parameter.*"

Please note that Sections 4.1 and 4.2 have been completely rewritten in the revised manuscript to fit with the results of the new simulations.

Kamath, H.G., Singh, M., Malviya, N. et al. GLObal Building heights for Urban Studies (UT-GLOBUS) for city- and street- scale urban simulations: Development and first applications. Sci Data 11, 886 (2024). https://doi.org/10.1038/s41597-024-03719-w

---

## Author Comment (AC6)

**Reply on RC2**

Replies to the Reviewers' remarks

We thank all the Reviewers for the careful revisions and the constructive comments to our work. All suggestions have been appreciated and have been taken into careful consideration in our revision. Below, we provide a detailed reply to all points raised by the reviewers.

**Referee: 2**

1) Is it a good assumption to not let longwave radiation interact with the tree canopy processes? Longwave radiation should be absorbed, transmitted, and emitted from the tree canopy like all other structures, which would then affect the radiative temperature and therefore heat flux partitioning. Please clarify what you mean by "The canopy interacts only with short-wave radiation and does not affect long-wave radiation components." on line 187

**R**: We fully agree with the reviewer's observation regarding the role of longwave radiation in tree canopy processes. Since the main objective of our work is to develop an efficient and lightweight modelling tool, we chose not to describe long-wave interactions with the tree canopy.
Nonetheless, we acknowledge the importance of this aspect, and additional clarification has been included in the manuscript in lines 195-199 as follows:

"*The interaction of the canopy with radiation is limited to shortwave components, as a modelling simplification. In the current version, longwave interactions with the tree canopy are neglected. This includes the (computationally expensive) omission of longwave reflection and exchange between multi-layer 2D assemblages of buildings, roads, and tree foliage. Remarkably, the observed thermal radiation fluxes are accurately reproduced despite this simplification.*"

2) Why use an empirical formulation for the partitioning of heat fluxes dependent on the shortwave radiation? Do street trees not have their own soil moisture stores that are similar to the street canyon gardens? Not taking into account the changes in soil moisture induced by urban trees, and the reduction in latent heat and subsequent changes to sensible and ground heat fluxes, could be biasing the results of this study (e.g. a reason why there are such large discrepancies in the latent heat results).

**R**: In the proposed model, street trees do not have a dedicated soil moisture reservoir. Instead, the transpiration process is explicitly represented as a function of downward short-wave radiation and this approach does not account for dynamic changes in soil moisture beneath trees.
We recognize this as a limitation of the current model version, and ongoing work aims at including BEP-Tree (Krayenhoff et al. 2020) in the model for a more complete representation of the role of vegetation. However, in most densely built urban areas, latent heat flux represents a relatively minor component of the surface energy budget. It is worth noting that, in the updated simulations, the latent heat flux values are consistent with those obtained from other modelling approaches.

A clarification has been added to the manuscript in lines 180-183 to explain this assumption and its implications more clearly.

"*In the current configuration, street trees are not assigned a dedicated soil moisture reservoir, limiting the representation of tree-induced soil moisture dynamics and potentially introducing biases in the partitioning of turbulent heat fluxes. On the other hand, in most dense urban areas, the latent heat is a small component of the surface energy budget.*"

3) Do you believe that the LCZ6 parameters you chose are representative for this space? Did you check the albedo and emissivity against remotely sensed averages? To my knowledge, LCZ give a range of values to select from, but these are likely to change given the age and type of architecture chosen for the study region.

**R**: We agree with the reviewer's observation regarding the representativeness of the LCZ6 parameters and the importance of validating albedo and emissivity values against remotely sensed data. However, in order to maintain internal consistency across the three different experiments presented in the paper and to align with the standardized framework proposed by the Urban-PLUMBER project, we chose to adopt the same set of LCZ6 parameter values across all experiments.

This decision was made to ensure comparability between experiments and to stay consistent with the reference configurations defined in the Urban-PLUMBER protocol. We acknowledge that some variability exists depending on the local architecture's age and typology, and this is indeed an important consideration for future model refinements.

Additional clarification has been added in the text in lines 261-263 to reflect this rationale.

"*To ensure internal consistency across experiments and alignment with the Urban-PLUMBER protocol, a standardized set of LCZ6 parameters was adopted, enabling compatibility while acknowledging the need for future refinements to capture local variability.*"

4) Upon reading the street canyon gardens section, I think that clarification is needed to discuss whether street trees are treated similarly (e.g. using ecohydrologic principals) or not.

**R**: Street trees are not treated in the same way as gardens.
Please note the clarification within section 2.2.2, reported in the revised manuscript in lines 180-182 as follows:

"*In the current configuration, street trees are not assigned a dedicated soil moisture reservoir, limiting the representation of tree-induced soil moisture dynamics and potentially introducing biases in the partitioning of turbulent heat fluxes.*"

5) I think that a more detailed investigation of the "why" behind sensible and latent differences in the single layer BEP BEM model is needed. For instance, why is the baseline doing better in the sensible heat flux compared to the more detailed versions of the model presented? Is there a lack of sensitivity/too much sensitivity to the parameters that were introduced? For the latent heat flux, it is tricky to tell what is going on without a better explanation of how green areas are represented. Is there soil moisture/ hydrology being simulated? Or is this the ratio

that was mentioned in the methods section? Examining the code shows that partitioning between sensible and latent heat fluxes for trees, which would be the major contributor to the latent heat signal, is using this ratio. More justification is needed on why this is appropriate given the biases that it introduces, given that urban trees do increase latent heat fluxes to be higher than those shown (even in modeling experiments, like related work with BEP-Tree (https://doi.org/10.1016/j.uclim.2020.100590) or tiling approaches in Noah-MP HUE that represent ecohydrology (https://doi.org/10.1029/2023WR035511)).

**R**: The results of the "baseline" experiment are broadly consistent with those of the "detailed" and "complex" experiments; however, it exhibits greater variability and an elevated cRMSE. This difference in performance is a consequence of variations in urban geometry. According to the LCZ classification, the "baseline" features a more compact and lower urban structure. This arrangement of buildings leads to less heat accumulation and release, resulting in lower BIAS for sensible heat fluxes.

Green areas are not explicitly treated according to the tree canopy cover ratio. Indeed, as described in Section 2.2.2, soil moisture and hydrological processes are represented, including a soil layer where moisture content evolves dynamically based on surface energy balance and precipitation inputs.

Regarding latent heat flux and its representation, we recognize this as a limitation of the current model version, and ongoing work aims at including BEP-Tree (Krayenhoff et al. 2020) in the model for a complete representation of the role of vegetation. However, despite the use of simplified descriptions of street trees, the model yields results that are consistent with those from similar parameterizations. This demonstrates the robustness of the approach, even when driven by relatively coarse urban input data.

Further discussions have been included in the revised manuscript in lines 444-447:

"*The "baseline" experiment shows a mixed behaviour in reproducing sensible heat fluxes ($Q_h$), with reduced BIAS but overestimated variability compared to the more detailed experiments. This is mainly attributed to its representation of a compact and low urban structure derived from the LCZ classification, which reduces heat accumulation and release dynamics.*"

6) Authors could give a more detailed breakdown of what they hypothesize is going wrong than "whose cause deserves further investigations, are present for latent heat flux (Qle)." As stated on line 414.

**R**: We acknowledge the lack of detail in the original version of the manuscript regarding this point. Please note that the revised version of the manuscript has been adapted to fit the new results. However, in response to the reviewer's suggestion, we have included additional and more specific details in lines 448-453 in the revised version of the manuscript.

"*The discrepancies observed in both sensible ($Q_h$) and latent heat flux ($Q_{le}$) is probably attributable to the limitations of the current model setup, which omits the presence of trees in green areas. As a result, the model does not account for the full extent of vegetative cover, particularly in areas with dense greenery. In addition, street trees may receive limited solar radiation due to shading effects and potential inaccuracies in urban geometry representation, which can further reduce their transpiration capacity. The absence of a dedicated soil moisture*"

*reservoir for trees also limits the simulation of their evapotranspirative contribution. These factors can contribute to an imbalance in the surface energy budget.*"

7) What is going on with the ground heat flux? I am assuming there are no measurements, but do the results from this flux look believable? I would think that because of the errors in Qh and Qle, Qg would be also too high, and thus could cause a warming feedback when introduced into weather or climate models like suggested in the discussion.

**R**: The Reviewer is correct in noting that no measurements of ground heat flux ($Q_g$) are available, which poses a challenge for directly evaluating the accuracy of this component. Nonetheless, we recognize the limitations of this approach and have clearly stated in the manuscript that the lack of observational data for $Q_g$ represents a constraint in evaluating model performance for this specific flux.
This limitation has been explicitly addressed in the revised text in lines 453-455.

"*The absence of observational data for ground heat flux ($Q_g$) prevents direct validation, limiting the evaluation of model performance for this component and warranting caution in interpreting its impact in coupled simulations.*"

8) The final point of this paper, "Further research includes experiments forcing the MLUCM BEP+BEM model with the ERA5 reanalysis to assess its sensitivity to various input parameters, including urban morphology and vegetation characteristics. Moreover, the model will be forced with climate projections to investigate the impact of climate change on the different urban processes, such as overheating, building energy demands, outdoor thermal comfort, and the efficacy of adaptation strategies, including urban greening, green and cool roofs, photovoltaic panels and hybrid sustainable infrastructure." is a lofty goal, and I would agree that this model is able to look at urban morphology pretty neatly. The issue is coming from the vegetation characteristics, urban greening, green and cool roofs, and the interactions with urban comfort and other applications. As of right now, the latent heat flux and sensible heat fluxes are wrong, which would then cause these to be erroneous. This model is on the right track, but there needs to be more justification/investigation/discussion on how we could use this model to look at urban climate adaptation with the errors that are present within the model right now.

**R**: We appreciate the reviewer's recognition of the model's potential in representing urban morphology and we fully agree that the future applications outlined in the paper are ambitious. As part of our ongoing development efforts, we acknowledge the need to further improve the representation of vegetation and the associated hydrological processes.
While on one hand, it is important to acknowledge the current limitations of the model, particularly in the representation of vegetation processes and the discrepancies observed in thermal fluxes, on the other hand, as demonstrated in our results, the model's performance is in line with or in some cases superior to other similar urban parameterizations.
Our primary objective has been to develop a computationally efficient tool capable of representing key urban processes, with a special focus on morphology and its influence on the urban energy balance assessing the effectiveness of urban mitigation strategies, such as greening or cool roofs, over broader spatial and temporal scales. We believe that the inaccuracies of the model do not prevent achieving a useful estimate of long term trends of

projections at the urban scale and their uncertainties, initiating to provide some crucial information that is requested by city planners and stakeholders.

In response to the reviewer's suggestion, we have revised and expanded the discussion in lines 469-474 in the manuscript to reflect these important considerations and to clearly communicate both the strengths and limitations of the current model framework.

"*Although the current model adopts simplified representations of vegetation and hydrological processes, the results demonstrate good agreement with, and in some cases outperform, other comparable urban parameterizations. These findings suggest that the model can provide a reasonable representation of key urban climate dynamics, even when driven by limited input detail. Its computational efficiency makes it particularly suited for exploring long-term trends and assessing large-scale mitigation strategies. Therefore, the framework represents a promising tool for supporting urban-scale climate analyses and informing decision-making processes.*"

9) When investigating the model code, the code is clean but there are some missed opportunities to give an indication of what each of the subroutines are doing. Consider adding those so that folks who want to add/modify this code base will know what is going on in each routine and call to the routine!

**R**: We agree that providing clearer documentation within the code is essential to support users who wish to understand, modify, or extend/implement the model. To address this, we will prepare a user guide that outlines how to properly set up and use the model, including descriptions of its main components and configuration options. In addition, we plan to release a second version of the code that includes more detailed and specific comments within each subroutine, as suggested. This will improve transparency and usability, making the model more accessible to the broader research community.

Minor points:
1. Line 59: missing a space between "1 Dimensional"

**R**: The error has been fixed

General comment for Review 2
The Reviewers' comments offered an opportunity to conduct a more in-depth investigation into the representation of latent heat flux in our model. This led to the discovery of a significant error: the tree and garden area fractions used in the three experiments were only 55% and 76%, respectively, of the correct values. This discrepancy systematically underestimated both the magnitude and variability of the latent heat flux.

Furthermore, the recently proposed UT-GLOBUS database by Kamath et al. (2024) was employed to enhance the representation of building distributions in the "complex" experimental setup, producing a generalized increase of the height of buildings. These two changes have led to a reduction of the negative latent heat bias of our former simulations. The updated simulations yield results that are consistent with those obtained from similar parameterization schemes.

Additional clarification has been included in the manuscript in lines 277-278 as follows:
"*For the "complex" experiment, the UT-GLOBUS database (Kamath et al., 2024) was employed to derive this parameter.*"

Please note that Sections 4.1 and 4.2 have been completely rewritten in the revised manuscript to fit with the results of the new simulations.

Kamath, H.G., Singh, M., Malviya, N. et al. GLObal Building heights for Urban Studies (UT-GLOBUS) for city- and street- scale urban simulations: Development and first applications. Sci Data 11, 886 (2024). https://doi.org/10.1038/s41597-024-03719-w

---

## Referee Report (RR1)

Reviewer Comments

MLUCM BEP+BEM: An offline one-dimensional Multi-Layer Urban Canopy Model based on the BEP+BEM scheme

The authors have done quite a lot of work in responding to the reviewers comments from the first round. The manuscript now does a better job of highlighting assumptions that are within this model formulation. I also appreciate the extra care they have done in responding to my comments about clarifying the performance of the energy fluxes.

After carefully reading through this new paper, I still have a few reservations about the presentation of this model. Specifically, this model is being presented to "bridge the mesoscale and microscale phenomena occurring in the planetary boundary layer and within the urban canopy, accounting for exchanges and feedback between different scales and processes" (line 63). Yet, there are critical processes that are missing that make this a tool that would not be useful outside of a very small subset of heavily urbanized regions that lack much vegetation. While those are important (the most heavily urbanized and likely to have the most intense impacts of heat), this model formulation is likely going to be severely biased due to the lack of hydrology and reliance on empirical formulations.

I agree that this model will be useful as either a quick analysis of longterm simulations of modulation of thermal parameters or after substantial model development be able to fill a gap in actionable science for climate adaptation that could be used by decision makers. Unfortunately, at the current state, the latter is not possible despite some language used in the manuscript. My comments are directed to help clarify this point and give a better representation of what this model could provide and where it would be helpful to use.

Comments (all are pretty major):

- As the authors have mentioned in their reviewer comments, a user manual should be created to be ready for publication when this paper is live to ensure that the code is as accessible as possible.
- Do the authors believe that 8 cm is enough hydrologically active soil to be able to model rain gardens? While appropriate for a green roof, specifically an extensive green roof with short shrubbery as is modeled by Zonato et al. 2021, rain gardens usually do not have such a shallow growing media and to not have an impermeable bottom. This could be a difference that is occurring due to terminology, where a rain garden could mean a planter boxes that have an impermeable bottom, but should thus be defined more clearly.

- I believe that the presentation of the results, specifically for sensible and latent heat, are obfuscating the real impact that these model simplifications (e.g. lack of hydrology and reliance on empirical coefficents) are causing. I would ask that the authors re-create Figure 3, but only during daylight hours. This would give a better idea of Sensible and Latent heat flux biases. Both fluxes are more variable during the daytime (latent heat is ~0 during the night most of the time, and sensible heat is slightly negative). This does not need to be in the main body of the text but should be pointed to for a clear representation that model structural development choices are creating.

- The new results, even after the new parameters that the authors have identified, do not introduce much model sensitivity. The improvement of the Baseline simulation compared to the Complex simulation are not as large as the "Results show that the integration of detailed, site-specific information on urban elements such as building geometry and vegetation generally improves the simulation of energy fluxes" on line 461 mentions. Please revise, especially as the changes between comparisons in Table 3 and Taylor diagrams in section 4.3 are not that large.

- Line 473: "Its computational efficiency makes it particularly suited for exploring long-term trends and assessing large-scale mitigation strategies." Please clarify what mitigation strategies that this model would be helpful in. The model, as is currently stands, would be useful in modulating radiative parameters, but the lack of hydrologic treatment and therefor the increase/decrease of latent heat would make this difficult to use in the widespread application of green infrastructure. To work with green infrastructure, one would need to add hydrology to the land surface model in a more sophisticated way. The authors may consider citing alternatives that would be appropriate for green infrastructure strategies when they clarify this point.

- Finally, the authors mention the computational efficiency as a key selling point for this model. It would help if the authors provided somewhere (could be in a table, could be in an SI figure) differences in computational load that was needed to run this model vs. the other models in this paper. Including runtime, number of CPUs, etc. would help justify the computational efficiency point in this paper.

---

## Author Response (AR2)

Review of egusphere-2025-219_R2

"MLUCM BEP+BEM: An offline one-dimensional Multi-Layer Urban Canopy Model based on the BEP+BEM Scheme"

by Gianluca Pappaccogli, Andrea Zonato, Alberto Martilli, Riccardo Buccolieri, and Piero Lionello

Replies to the Reviewers' remarks

We sincerely thank the Reviewer for the thorough evaluation and constructive feedback. The comments have been greatly appreciated and have informed a careful revision of the manuscript. Below, we provide detailed responses to each point raised.

**Referee: 1**

The authors have done quite a lot of work in responding to the reviewers comments from the first round. The manuscript now does a better job of highlighting assumptions that are within this model formulation. I also appreciate the extra care they have done in responding to my comments about clarifying the performance of the energy fluxes.
After carefully reading through this new paper, I still have a few reservations about the presentation of this model. Specifically, this model is being presented to "bridge the mesoscale and microscale phenomena occurring in the planetary boundary layer and within the urban canopy, accounting for exchanges and feedback between different scales and processes" (line 63). Yet, there are critical processes that are missing that make this a tool that would not be useful outside of a very small subset of heavily urbanized regions that lack much vegetation. While those are important (the most heavily urbanized and likely to have the most intense impacts of heat), this model formulation is likely going to be severely biased due to the lack of hydrology and reliance on empirical formulations.

I agree that this model will be useful as either a quick analysis of longterm simulations of modulation of thermal parameters or after substantial model development be able to fill a gap in actionable science for climate adaptation that could be used by decision makers. Unfortunately, at the current state, the latter is not possible despite some language used in the manuscript. My comments are directed to help clarify this point and give a better representation of what this model could provide and where it would be helpful to use.

We understand the concerns regarding the model's current limitations, particularly the absence of an explicit hydrological component and the use of empirical formulations. In the revised manuscript, we have clarified the intended scope and applicability of the model, emphasizing its focus on densely urbanized areas with limited vegetation cover. We also acknowledge that further development will be needed to extend its applicability to a broader range of urban contexts. The suggested clarifications have been implemented to better reflect the model's current capabilities and potential future developments. If on one hand the MLUCM BEP+BEM model does not currently include a fully developed hydrological component, it offers a highly detailed representation of turbulence and building-atmosphere energy exchanges, features that are often simplified or absent in other comparable models. Our results indicate that the performance of MLUCM BEP+BEM is comparable to, and in several key variables even exceeds, that of other state-of-the-art models.

Comments
1) As the authors have mentioned in their reviewer comments, a user manual should be created to be ready for publication when this paper is live to ensure that the code is as accessible as possible.

**R**: We thank the reviewer for emphasizing the importance of accessibility and usability of the proposed model. We have prepared a user manual to accompany the first release of the model (please find the file in Zenodo https://doi.org/10.5281/zenodo.14773142). This manual is intended to support users in setting up and running simulations with the current version of MLUCM BEP+BEM model. Furthermore, the authors are committed to ensuring that the model becomes as accessible and widely usable as possible. Future efforts will focus on enhancing documentation, facilitating user support, and promoting broader adoption within the research community.

2) Do the authors believe that 8 cm is enough hydrologically active soil to be able to model rain gardens? While appropriate for a green roof, specifically an extensive green roof with short shrubbery as is modeled by Zonato et al. 2021, rain gardens usually do not have such a shallow growing media and to not have an impermeable bottom. This could be a difference that is occurring due to terminology, where a rain garden could mean a planter boxes that have an impermeable bottom, but should thus be defined more clearly.

**R**: We fully agree that the hydrological characteristics of rain gardens typically involve deeper soil layers and often permeable bottoms, which differ significantly from the active soil assumed in our current setup. These limitations are acknowledged and will be addressed in future developments of the model.
We have clarified this point in the revised manuscript in lines 217-220.

"*The modeled gardens are implemented with a simplified soil layer structure and a bottom boundary condition that does not account for infiltration into deeper permeable soil. While this setup may resemble planter boxes with impermeable bottoms, it does not aim to capture the full range of hydrological responses of gardens to precipitation and complex evapotranspiration processes.*"

Furthermore, a note is included in the user manual to warn about the limitations of the model to represent the effect of large gardens.

3) I believe that the presentation of the results, specifically for sensible and latent heat, are obfuscating the real impact that these model simplifications (e.g. lack of hydrology and reliance on empirical coefficients) are causing. I would ask that the authors re-create Figure 3, but only during daylight hours. This would give a better idea of Sensible and Latent heat flux biases. Both fluxes are more variable during the daytime (latent heat is ~0 during the night most of the time, and sensible heat is slightly negative). This does not need to be in the main body of the text but should be pointed to for a clear representation that model structural development choices are creating.

**R**: In the present results we follow the Urban-PLUMBER protocol, to allow the comparison with other models involved in the experiment, as well as to provide publicly available output. This protocol does not consider the division between day and night. However, we thank the

Reviewer for the valuable suggestion, and we carried out the requested analysis. We added in the supplementary material a new version of the Figure 3 considering only daylight hours (i.e. shortwave downward radiation > 10 W m$^{-2}$) and added to the article (lines 356-363) that: "*Further analysis indicates that the model reproduces observations more accurately during the daytime than at night (Fig. S1). The overall unsatisfactory performance of the model appears to be primarily due to the unrealistic simulation of nighttime fluxes, whereas daytime fluxes are reasonably well captured. As a result, the sensible heat flux is well reproduced when it represents a significant component of the surface energy budget, and less accurately reproduced at night, when its contribution is minimal.*

*This discrepancy is not expected to significantly affect the estimation of quantities such as building energy consumption, which is in the focus of the model scope. Similar considerations apply to the latent heat flux, albeit to a lesser extent. These findings are consistent with the widely recognized limited role of latent heat fluxes in densely urbanized environments.*"

The version of Figure 3 considering only daytime is provided here (Fig 1R2) and it is added to the supplementary material of the manuscript (i.e. Fig. S1), while Table 1R2 reports the performance metrics.

[Figure]

Figure 1R2: As in Figure 3, but during daytime (i.e. shortwave downward radiation > 10 W m$^{-2}$).

|  | Qh | Qle |
|---|---|---|
| BIAS | 15.98 | -1.19 |
| NME | 0.33 | 0.62 |
| SLOPE | 1.13 | 0.55 |
| COR | 0.92 | 0.58 |

Table 1R2 Statistics of the MLUCM BEP+BEM model for "complex" experiment during daytime.

4) The new results, even after the new parameters that the authors have identified, do not introduce much model sensitivity. The improvement of the Baseline simulation compared to the Complex simulation are not as large as the "Results show that the integration of detailed, site-specific information on urban elements such as building geometry and vegetation generally improves the simulation of energy fluxes" on line 461 mentions. Please revise, especially as the changes between comparisons in Table 3 and Taylor diagrams in section 4.3 are not that large.

**R**: We agree with the Reviewer. We have revised the sentence at lines 482-483 to better reflect the results .

*"Results show that the integration of detailed, site-specific information on urban elements such as building geometry and vegetation lead to some improvements in terms of correlation, cRMSE, and STD in the simulation of latent and sensible heat fluxes."*

5) Line 473: "Its computational efficiency makes it particularly suited for exploring long-term trends and assessing large-scale mitigation strategies." Please clarify what mitigation strategies that this model would be helpful in. The model, as is currently stands, would be useful in modulating radiative parameters, but the lack of hydrologic treatment and therefor the increase/decrease of latent heat would make this difficult to use in the widespread application of green infrastructure. To work with green infrastructure, one would need to add hydrology to the land surface model in a more sophisticated way. The authors may consider citing alternatives that would be appropriate for green infrastructure strategies when they clarify this point.

**R**: We have revised the text to clarify the types of mitigation strategies for which the model is currently best suited in lines 494-499. In particular, we acknowledge the limitations that arise when simulating green infrastructure interventions, such as extensive greening or the planting of trees due to the model's reliance on simplified approximations. These can lead to imbalances in the representation of energy fluxes, especially in highly vegetated urban areas, given the absence of an explicit hydrological component.
This point has now been addressed in the revised manuscript, where we specify that while the model may not be ideal for fully capturing the impacts of green infrastructure, it is well structured to assess changes in the thermal and radiative properties of buildings, including interventions such as green roofs, photovoltaic panels, or their combination. These features make the model particularly suitable for evaluating long-term trends and energy-efficiency-oriented mitigation strategies at the urban scale.

*"Its computational efficiency makes it particularly suited for exploring long-term trends and assessing mitigation strategies focused on the thermal and radiative properties of the built environment, such as the implementation of green and cool roofs, photovoltaic panels, or energy retrofitting measures. Conversely, care should be taken when using the model to evaluate the role of gardens and street trees, due to the simplified treatment of soil hydrology. In such cases, more detailed models such as the Urban Tethys-Chloris (UT&C) model (Meili et al., 2020), which explicitly account for ecohydrological processes, may provide a more accurate representation of vegetation effects on urban climate."*

*Meili, N., Manoli, G., Burlando, P., Bou-Zeid, E., Chow, W. T. L., Coutts, A. M., Daly, E., Nice, K. A., Roth, M., Tapper, N. J., Velasco, E., Vivoni, E. R., and Fatichi, S.: An urban ecohydrological model to quantify the effect of vegetation on urban climate and hydrology (UT&C v1.0), Geosci. Model Dev., 13, 335–362, https://doi.org/10.5194/gmd-13-335-2020, 2020.*

6) Finally, the authors mention the computational efficiency as a key selling point for this model. It would help if the authors provided somewhere (could be in a table, could be in an SI figure) differences in computational load that was needed to run this model vs. the other models in this paper. Including runtime, number of CPUs, etc. would help justify the computational efficiency point in this paper.

**R**: In the manuscript, we refer to the computational efficiency of the proposed column model primarily in relation to the BEP+BEM scheme when coupled online with a full mesoscale model such as Weather Research and Forecasting (WRF) model. However, it is important to note that direct comparisons of computational time can be highly dependent on the specific model configuration, the nature of the application, and the characteristics of the computing system used. For this reason, we have provided in the revised manuscript (lines 303-309) the simulation times and technical specifications of the workstation used for the case study presented in this work, to offer a clear reference framework for evaluating computational performance.

"*The model runs at one-minute time steps, with an average computational speed of approximately 4–5 ms per time step. A typical simulation covering one year of data requires approximately 30–40 minutes on a workstation using one Intel® Xeon® Gold 5218 CPU @ 2.30GHz with 2 GB RAM, operating in a virtualized environment (VMware). A fully coupled mesoscale model (e.g., WRF with BEP+BEM; Vidal et al., 2021), typically requires more than one day for a 24 hours simulation using a single core (decreasing to 1 hour in a 64-core computer). Though computational costs may vary depending on the specific application and the hardware used, it is clear that MLUCM BEP+BEM offers an enormously reduced computational cost, enabling faster simulations and making it particularly suitable for long-term studies.*"

Vidal, V., Cortés, A., Badia, A., Villalba, G. (2021). Evaluating WRF-BEP/BEM Performance: On the Way to Analyze Urban Air Quality at High Resolution Using WRF-Chem+BEP/BEM. In: Paszynski, M., Kranzlmüller, D., Krzhizhanovskaya, V.V., Dongarra, J.J., Sloot, P.M. (eds) Computational Science – ICCS 2021. ICCS 2021. Lecture Notes in Computer Science(), vol 12746. Springer, Cham. https://doi.org/10.1007/978-3-030-77977-1_41